# Online Minimax Multiobjective Optimization: Multicalibeating and Other Applications

**Daniel Lee[1], Georgy Noarov[1], Mallesh Pai[2], Aaron Roth[1]**
[1] University of Pennsylvania, [2] Rice University
daniellee@alumni.upenn.edu, gnoarov@seas.upenn.edu,
mallesh.pai@rice.edu, aaroth@cis.upenn.edu

## Abstract

We introduce a simple but general online learning framework in which a learner plays against an adversary in a vector-valued game that changes every round. Even though the learner's objective is not convex-concave (and so the minimax theorem does not apply), we give a simple algorithm that can compete with the setting in which the adversary must announce their action first, with optimally diminishing regret. We demonstrate the power of our framework by using it to (re)derive optimal bounds and efficient algorithms across a variety of domains, ranging from multicalibration to a large set of no regret algorithms, to a variant of Blackwell's approachability theorem for polytopes with fast convergence rates. As a new application, we show how to "(multi)calibeat" an arbitrary collection of forecasters — achieving an exponentially improved dependence on the number of models we are competing against, compared to prior work.

## 1 Introduction

We introduce and study a simple but powerful framework for online adversarial multiobjective minimax optimization. At each round $t$, an adaptive adversary chooses an environment for the learner to play in, defined by a convex compact action set $\mathcal{X}^t$ for the learner, a convex compact action set $\mathcal{Y}^t$ for the adversary, and a $d$-dimensional continuous loss function $\ell^t : \mathcal{X}^t \times \mathcal{Y}^t \to [-1, 1]^d$ that, in each coordinate, is convex in the learner's action and concave in the adversary's action. The learner then chooses an action, or distribution over actions, $x^t$, and the adversary responds with an action $y^t$. This results in a loss vector $\ell^t(x^t, y^t)$, which accumulates over time. The learner's goal is to minimize the maximum accumulated loss over each of the $d$ dimensions: $\max_{j \in [d]} \left( \sum_{t=1}^{T} \ell_j^t(x^t, y^t) \right)$.

One may view the environment chosen at each round $t$ as defining a zero-sum game in which the learner wishes to minimize the *maximum* coordinate of the resulting loss vector. The objective of the learner in the stage game in isolation can be written as:[1]

$$w_L^t = \inf_{x^t \in \mathcal{X}^t} \max_{y^t \in \mathcal{Y}^t} \left( \max_{j \in [d]} \ell_j^t(x^t, y^t) \right).$$

Unfortunately, although $\ell_j^t$ is convex-concave in each coordinate, the maximum over coordinates does not preserve concavity for the adversary. Thus the minimax theorem does not hold, and the value of the game in which the learner moves first (defined above) is larger than the value of the game in which the adversary moves first— that is, $w_L^t > w_A^T$, where $w_A^t$ is defined as:

---

[1] A brief aside about the "inf max max" structure of $w_L^t$: since each $\ell_j^t$ is continuous, so is $\max_j \ell_j^t$, and hence $\max_y (\max_j \ell_j^t)$ is attained on the compact set $\mathcal{Y}^t$. However, $\max_y (\max_j \ell_j^t)$ may not be a continuous function of $x$ and therefore the infimum over $\mathcal{X}^t$ need not be attained.

36th Conference on Neural Information Processing Systems (NeurIPS 2022).

$$w_A^t = \sup_{y^t \in \mathcal{Y}^t} \min_{x^t \in \mathcal{X}^t} \left( \max_{j \in [d]} \ell_j^t(x^t, y^t) \right).$$

Nevertheless, fixing a series of $T$ environments chosen by the adversary, this defines in hindsight an aspirational quantity $W_A^T = \sum_{t=1}^T w_A^t$, summing the adversary-moves-first value of the constituent zero sum games. Despite the fact that these values are not individually obtainable in the stage games, we show that they are approachable on average over a sequence of rounds, i.e., there is an algorithm for the learner that guarantees that against any adversary,

$$\max_{j \in [d]} \left( \frac{1}{T} \sum_{t=1}^T \ell_j^t(x^t, y^t) \right) \leq \frac{1}{T} W_A^T + 4\sqrt{\frac{2 \ln d}{T}}.$$

Our derivation is elementary and based on a minimax argument, and is a development of a game-theoretic argument from the calibration literature due to Hart [2020] and Fudenberg and Levine [1999].[2] The generic algorithm plays actions at every round $t$ according to a minimax equilibrium strategy in a surrogate game that is derived both from the environment chosen by the adversary at round $t$, as well as from the history of play so far on previous rounds $t' < t$. The loss in the surrogate game is convex-concave (and so we may apply minimax arguments), and can be used to upper bound the loss in the original games.

We then show that this simple framework can be instantiated to derive a wide array of optimal bounds, and that the corresponding algorithms can be derived in closed form by solving for the minimax equilibrium of the corresponding surrogate game. Despite its simplicity, our framework has a number of applications to online learning— we sketch these below.

**"Multi-Calibeating":** Foster and Hart [2021] recently introduced the notion of "calibeating" an arbitrary online forecaster: making online calibrated predictions about an adversarially chosen sequence of inputs that are guaranteed to have lower squared error than an arbitrary predictor $f$, where the improvement in error approaches $f$'s calibration error in hindsight. Foster and Hart give two methods for calibeating an arbitrary collection of predictors $\mathcal{F}$ simultaneously, but these methods have an exponential and polynomial dependence in their convergence bounds on $|\mathcal{F}|$, respectively.

Using our framework, we can derive optimal online bounds for online *multicalibration* [Hébert-Johnson et al., 2018, Gupta et al., 2022], and as an application, obtain bounds for calibeating arbitrary collection of models with only a *logarithmic* dependence on $|\mathcal{F}|$. Our algorithm naturally extends to the more general problem of online "multi-calibeating" — i.e. combining the goals of online multicalibration and calibeating. Namely, we give an algorithm for making real-valued predictions given contexts from some space $\Theta$. The algorithm is parameterized by (i) a collection $\mathcal{G} \subseteq 2^\Theta$ of (arbitrary, potentially intersecting) subsets of $\Theta$ that we might envision to represent e.g. different demographic groups in a setting in which we are making predictions about people; and (ii) an arbitrary collection of predictors $\mathcal{F}$. We promise that our predictions are calibrated not just overall, but simultaneously within each group $g \in \mathcal{G}$ — and moreover, that we *calibeat* each predictor $f \in \mathcal{F}$ not just overall, but simultaneously within each group $g \in \mathcal{G}$. We do this by proving an online analogue of what Hébert-Johnson et al. [2018] call a "do no harm" property in the batch setting using a similar technique: multicalibrating with respect to the level sets of the predictors.

**Fast Polytope Blackwell Approachability:** We give a variant of Blackwell's Approachability Theorem [Blackwell, 1956] for approaching a polytope. Standard methods approach a set in Euclidean distance, at a rate polynomial in the payoff dimension. In contrast, we give a dimension-independent approachability guarantee: we approximately satisfy all halfspace constraints defining the polytope, after *logarithmically* many rounds in the number of constraints, a significant improvement over a polynomial dimensional dependence in many settings. It is equivalent to the results of Perchet [2015], which show that the negative orthant $\mathbb{R}_{\leq 0}^d$ is approachable in the $\ell_\infty$ metric with a $\log(d)$ dependence in the convergence rate. This result follows immediately from a specialization of our framework that does not require changing the environment at each round, highlighting the connection between our framework and approachability. We remark that approachability has been extended in a number of ways in recent years [Mannor et al., 2014a,b, Perchet and Mannor, 2013]. However most of our other applications take advantage of the flexibility of our framework to play a different game at each

[2]This argument was extended in Gupta et al. [2022] to obtain fast rates and explicit algorithms for multicalibration and multivalidity.

round (which can be defined by context) with potentially different action sets, and so do not directly follow from Blackwell approachability. Therefore, while many of our regret bounds could be derived from approachability to the negative orthant by enlarging the action space exponentially to simulate aspects of our framework, this approach would not easily lead to *efficient* algorithms.

**Recovering Expert Learning Bounds:**  Algorithms and optimal bounds for various expert learning problems fall naturally out of our framework as corollaries. This includes external regret [Vovk, 1990, Littlestone and Warmuth, 1994], internal and swap regret [Foster and Vohra, 1998, Hart and Mas-Colell, 2000, Blum and Mansour, 2007], adaptive regret [Littlestone and Warmuth, 1994, Hazan and Seshadhri, 2009, Adamskiy et al., 2012], sleeping experts [Freund et al., 1997, Blum, 1997, Blum and Mansour, 2007, Kleinberg et al., 2010], and the recently introduced multi-group regret [Blum and Lykouris, 2020, Rothblum and Yona, 2021]. Multi-group regret refers to a contextual prediction problem in which the learner gets contexts from $\Theta$ before each round. It is parameterized by a collection of groups $\mathcal{G} \subseteq 2^{\Theta}$: e.g., if the predictions concern people, $\mathcal{G}$ may represent an arbitrary, intersecting set of demographic groups. Here the "experts" are different models that make predictions on each instance; the goal is to attain no-regret not just overall, but also on the subset of rounds corresponding to contexts from each $g \in \mathcal{G}$. Multi-group regret, like multicalibration, is one of the few solution concepts in the algorithmic fairness literature known not to involve tradeoffs with overall accuracy [Globus-Harris et al., 2022]. Blum and Lykouris [2020] derived their algorithm for online multigroup regret via a reduction to sleeping experts, and Gupta et al. [2022] derived their algorithm for online multicalibration via a direct argument. Here we derive online algorithms for both multicalibration and multigroup regret as corollaries of the same fundamental framework.

# 2  General Framework

## 2.1  The Setting

A Learner (she) plays against an Adversary (he) over rounds $t \in [T] := \{1, \ldots, T\}$. Over these rounds, she accumulates a $d$-dimensional loss vector ($d \geq 1$), where each round's loss vector lies in $[-C, C]^d$ for some $C > 0$. At each round $t$, the Learner and the Adversary interact as follows:

1. Before round $t$, the Adversary selects and reveals to the Learner an *environment* comprising:

   (a) The Learner's and Adversary's respective convex compact action sets $\mathcal{X}^t, \mathcal{Y}^t$ embedded into a finite-dimensional Euclidean space;

   (b) A continuous vector loss function $\ell^t(\cdot, \cdot) : \mathcal{X}^t \times \mathcal{Y}^t \to [-C, C]^d$, with each $\ell_j^t(\cdot, \cdot) : \mathcal{X}^t \times \mathcal{Y}^t \to [-C, C]$ (for $j \in [d]$) convex in the 1st and concave in the 2nd argument.

2. The Learner selects some $x^t \in \mathcal{X}^t$.

3. The Adversary observes the Learner's selection $x^t$, and responds with some $y^t \in \mathcal{Y}^t$.

4. The Learner suffers (and observes) the loss vector $\ell^t(x^t, y^t)$.

The Learner's objective is to minimize the value of the maximum dimension of the accumulated loss vector after $T$ rounds—in other words, to minimize: $\max_{j \in [d]} \sum_{t \in [T]} \ell_j^t(x^t, y^t)$.

To benchmark the Learner's performance, we consider the following quantity at each round $t$:

**Definition 2.1** (The Adversary-Moves-First (AMF) Value at Round $t$). *The Adversary-Moves-First value of the game defined by the environment $(\mathcal{X}^t, \mathcal{Y}^t, \ell^t)$ at round $t$ is:*

$$w_A^t := \sup_{y^t \in \mathcal{Y}^t} \min_{x^t \in \mathcal{X}^t} \left( \max_{j \in [d]} \ell_j^t(x^t, y^t) \right).$$

If the Adversary had to reveal $y^t$ first and the Learner could best respond, $w_A^t$ would be the smallest value of the maximum coordinate of $\ell^t$ she could guarantee. However, the function $\max_{j \in [d]} \ell_j^t(x^t, y^t)$ is not convex-concave (as the max does not preserve concavity); hence the minimax theorem does not apply, making this value unobtainable for the Learner, who is in fact obligated to reveal $x^t$ first. However, we can define regret to a benchmark given by the cumulative AMF values of the games:

**Definition 2.2** (Adversary-Moves-First (AMF) Regret). *On transcript $\pi^t = \{(\mathcal{X}^s, \mathcal{Y}^s, \ell^s), x^s, y^s\}_{s=1}^t$, we define the Learner's Adversary Moves First (AMF) Regret for the $j^{th}$ dimension at time $t$ to be:*

$$R_j^t(\pi^t) := \sum_{s=1}^t \ell_j^s(x^s, y^s) - \sum_{s=1}^t w_A^s.$$

*The overall AMF Regret is then defined as follows: $R^t(\pi^t) = \max_{j \in [d]} R_j^t$.[3]*

Again, the game played at each round is *not* convex-concave, so we cannot get $R^T \le 0$. Instead, we will aim to obtain *sublinear* AMF regret, worst-case over adaptive adversaries: $R^T = o(T)$.

## 2.2 General Algorithm

Our algorithmic framework will be based on a natural idea: instead of directly grappling with the maximum coordinate of the cumulative vector valued loss, we upper bound the AMF regret with a one-dimensional "soft-max" surrogate loss function, which the algorithm will then aim to minimize.

**Definition 2.3** (Surrogate loss). *Fixing a parameter $\eta \in (0, 1)$, we define our surrogate loss function (that implicitly depends on the transcript $\pi^t$ through the respective round $t$) as:*

$$L^t := \sum_{j \in [d]} \exp\left(\eta R_j^t\right) \text{ for } t \in [T], \quad \text{and } L^0 := d.$$

This surrogate loss tightly bounds the AMF regret $R^T = \max_{j \in [d]} R_j^T$:

**Lemma 2.1.** *The Learner's AMF Regret is upper bounded using the surrogate loss as: $R^T \le \frac{\ln L^T}{\eta}$.*

Next we observe a simple but important bound on the per-round increase in the surrogate loss.

**Lemma 2.2.** *For any $t$, any transcript through round $t$, and any $\eta \le \frac{1}{2C}$, it holds that:*

$$L^t \le \left(4\eta^2 C^2 + 1\right) L^{t-1} + \eta \sum_{j \in [d]} \exp\left(\eta R_j^{t-1}\right) \cdot \left(\ell_j^t\left(x^t, y^t\right) - w_A^t\right).$$

The proof is very simple (see Appendix B.1): we write out the quantity $L^t - L^{t-1}$, use the definition of AMF regret $R^t$, and then bound $L^t - L^{t-1}$ via the inequality $e^x \le 1 + x + x^2$ for $|x| \le 1$.

We now exploit Lemma 2.2 to bound the final surrogate loss $L^T$ and obtain a game-theoretic algorithm for the Learner that attains this bound. While the above steps should remind the reader of a standard derivation of the celebrated Exponential Weights algorithm via bounding a log-sum-exp potential function, the next lemma is the novel ingredient that makes our framework significantly more general by relying on Sion's powerful generalization of the Minimax Theorem to convex-concave games.

**Lemma 2.3.** *For any $\eta \le \frac{1}{2C}$, the Learner can ensure that the final surrogate loss is bounded as:*

$$L^T \le d \left(4\eta^2 C^2 + 1\right)^T.$$

*Proof sketch; see Appendix B.1.* Define, for $t \in [T]$, continuous convex-concave functions $u^t : \mathcal{X}^t \times \mathcal{Y}^t \to \mathbb{R}$ by: $u^t(x, y) := \sum_{j \in [d]} \exp\left(\eta R_j^{t-1}\right)\left(\ell_j^t(x, y) - w_A^t\right)$. If the Learner can ensure $u^t(x^t, y^t) \le 0$ on all rounds $t \in [T]$ regardless of the Adversary's play, then Lemma 2.2 implies $L^t \le \left(4\eta^2 C^2 + 1\right) L^{t-1}$ for all $t \in [T]$, leading to the desired bound on $L^T$. Due to the continuous convex-concave nature of each $u^t$ (inherited from the loss coordinates $\ell_j^t$), we can apply Sion's Minimax Theorem to conclude that: $\min_{x^t \in \mathcal{X}^t} \max_{y^t \in \mathcal{Y}^t} u^t\left(x^t, y^t\right) = \max_{y^t \in \mathcal{Y}^t} \min_{x^t \in \mathcal{X}^t} u^t\left(x^t, y^t\right)$.

In words, the Learner has a so-called *minimax-optimal* strategy $x^t$, that achieves (worst-case over all $y^t \in \mathcal{Y}^t$) value $u^t(x^t, y^t)$ as low as if the Adversary moved first and the Learner could best-respond. But in the latter counterfactual scenario, using the definitions of $u^t$ and the Adversary-moves-first value $w_A^t$, we can easily see that by best-responding to the Adversary, the Learner would always guarantee herself value $\le 0$: that is, $\max_{y^t \in \mathcal{Y}^t} \min_{x^t \in \mathcal{X}^t} u^t\left(x^t, y^t\right) \le 0$. Thus, $\min_{x^t \in \mathcal{X}^t} \max_{y^t \in \mathcal{Y}^t} u^t\left(x^t, y^t\right)$, and so by playing minimax-optimally at every round $t \in [T]$, the Learner will guarantee $u^t\left(x^t, y^t\right) \le 0$ for all $t$, leading to the desired regret bound. $\square$

---

[3]We will generally elide the dependence on the transcript and simply write $R_j^t$ and $R^t$.

In fact, via a simple algebraic transformation (see Appendix B.1) taking advantage of the values $w_A^t$ being independent of the actions $x^t, y^t$, we can explicitly express the Learner's minimax optimal strategies at all rounds as: $\underset{x \in \mathcal{X}^t}{\operatorname{argmin}} \underset{y \in \mathcal{Y}^t}{\max} u^t(x, y) = \underset{x \in \mathcal{X}^t}{\operatorname{argmin}} \underset{y \in \mathcal{Y}^t}{\max} \sum_{j \in [d]} \frac{\exp\left(\eta \sum_{s=1}^{t-1} \ell_j^s(x^s, y^s)\right)}{\sum_{i \in [d]} \exp\left(\eta \sum_{s=1}^{t-1} \ell_i^s(x^s, y^s)\right)} \ell_j^t(x, y).$

Together with the proof of Lemma 2.3, this immediately gives the following algorithm for the Learner that achieves the desired bound on $L^T$ (and thus, as we will show, on the AMF regret $R^T$).

---

**Algorithm 1:** General Algorithm for the Learner that Achieves Sublinear AMF Regret

---

    **for** rounds $t = 1, \ldots, T$ **do**

        Learn adversarially chosen $\mathcal{X}^t, \mathcal{Y}^t$, and loss function $\ell^t(\cdot, \cdot)$.

        Let
$$\chi_j^t := \frac{\exp\left(\eta \sum_{s=1}^{t-1} \ell_j^s(x^s, y^s)\right)}{\sum_{i \in [d]} \exp\left(\eta \sum_{s=1}^{t-1} \ell_i^s(x^s, y^s)\right)} \text{ for } j \in [d].$$

        Play
$$x^t \in \underset{x \in \mathcal{X}^t}{\operatorname{argmin}} \underset{y \in \mathcal{Y}^t}{\max} \sum_{j \in [d]} \chi_j^t \cdot \ell_j^t(x, y).$$

        Observe the Adversary's selection of $y^t \in \mathcal{Y}^t$.

---

**Theorem 2.1** (AMF Regret guarantee of Algorithm 1). *For any $T \geq \ln d$, Algorithm 1 with learning rate $\eta = \sqrt{\frac{\ln d}{4TC^2}}$ obtains, against any Adversary, AMF regret bounded by: $R^T \leq 4C\sqrt{T \ln d}$.*

Indeed, using Lemma 2.1, then Lemma 2.3, then $1 + x \leq e^x$, and finally setting $\eta = \sqrt{\frac{\ln d}{4TC^2}}$, we get:

$$R^T \leq \frac{\ln L^T}{\eta} \leq \frac{\ln\left(d\left(4\eta^2 C^2 + 1\right)^T\right)}{\eta} \leq \frac{\ln\left(d \exp\left(4T\eta^2 C^2\right)\right)}{\eta} = \frac{\ln d}{\eta} + 4TC^2\eta = 4C\sqrt{T \ln d}.$$

**Remark 2.1.** *Our framework is easy to adapt to the setting where the Learner randomizes, at each round, amongst a finite set of actions $\mathcal{A}^t$ (i.e. $\mathcal{X}^t = \Delta\mathcal{A}^t$), and wishes to obtain in-expectation and high-probability AMF regret bounds. This is useful in all our applications below. Additionally, our AMF regret bounds are robust to the Learner playing only an approximate (rather than exact) minimax strategy at each round: we use this to derive our simple multicalibration algorithm below. See Appendix B.2 for both these extensions.*

## 3 Deriving No-X-Regret Algorithms from Our Framework

The core of our framework — the Adversary-Moves-First regret — is strictly more general than a very large variety of known regret notions including: *external, internal, swap, adaptive, sleeping-experts, multigroup, and wide-range ($\Phi$) regret*. Specifically, in Appendix C, we use our framework to derive simple $O(\sqrt{T})$-regret algorithms for what we call *subsequence regret*, which encapsulates all these regret forms. In each of these cases, our generic algorithm is efficient, and often specializes (by computing a minimax equilibrium strategy in closed form) to simple combinatorial algorithms that had been derived from first principles in prior work. We note that in any problem that involves context or changing action spaces (as the sleeping experts problem does), we are taking advantage of the flexibility of our framework to present a different environment at every round, which distinguishes our framework from more standard Blackwell approachability arguments. In fact, as we will see in Section 5 below, our framework recovers fast Blackwell approachability as a special case.

For our general subsequence regret algorithms, please see Appendix C. Now, as a warm-up application of our framework, we directly instantiate it for the simplest case of obtaining $O(\sqrt{T})$ *external* regret.

**Simple Learning From Expert Advice: External Regret** In the classical experts learning setting Littlestone and Warmuth [1994], the Learner has a set of pure actions ("experts") $\mathcal{A}$. At the outset of each round $t \in [T]$, the Learner chooses a distribution over experts $x^t \in \Delta\mathcal{A}$. The Adversary then comes up with a vector of losses $r^t = (r_a^t)_{a \in \mathcal{A}} \in [0, 1]^{\mathcal{A}}$ corresponding to each expert. Next, the Learner samples $a^t \sim x^t$, and experiences loss corresponding to the expert she chose: $r_{a^t}^t$. The Learner also gets to observe the entire vector of losses $r^t$ for that round. The goal of the

Learner is to achieve sublinear *external regret* — that is, to ensure that the difference between her cumulative loss and the loss of the best fixed expert in hindsight grows sublinearly with $T$: $R_{\text{ext}}^T(\pi^T) := \sum_{t \in [T]} r_{a^t}^t - \min_{j \in \mathcal{A}} \sum_{t \in [T]} r_j^t = o(T)$.

**Theorem 3.1.** *Fix a finite pure action set $\mathcal{A}$ for the Learner and a time horizon $T \geq \ln |\mathcal{A}|$. Then, an instantiation of our framework's Algorithm B.1 lets the Learner achieve the following regret bounds:*

$$\mathbb{E}_{\pi^T}\left[R_{\text{ext}}^T\left(\pi^T\right)\right] \leq 4\sqrt{T \ln |\mathcal{A}|}, \quad \text{and } R_{\text{ext}}^T\left(\pi^T\right) \leq 8\sqrt{T \ln \tfrac{|\mathcal{A}|}{\delta}} \text{ with prob. } 1 - \delta.$$

*Proof.* We instantiate (the probabilistic version of) our framework (see Section B.2.1).

At all rounds, the Learner's pure action set is $\mathcal{A}$, and the Adversary's strategy space is the convex and compact set $[0,1]^{|\mathcal{A}|}$, from which each round's collection $(r_a^t)_{a \in \mathcal{A}}$ of all actions' losses is selected. Next, we define a $|\mathcal{A}|$-dimensional loss function $\ell^t = (\ell_j^t)_{j \in \mathcal{A}}$, where each coordinate loss $\ell_j^t$ expresses the regret of the Learner's chosen action $a$ relative to action $j \in \mathcal{A}$:

$$\ell_j^t(a, r^t) = r_a^t - r_j^t, \quad \text{for } a \in \mathcal{A}, r^t \in [0,1]^{|\mathcal{A}|}.$$

By Theorem B.1, $\mathbb{E}\left[\max_{j \in \mathcal{A}} \sum_{t \in [T]} \ell_j^t(a^t, r^t) - \sum_{t \in [T]} w_A^t\right] \leq 4\sqrt{T \ln |\mathcal{A}|}$, where $w_A^t$ is the AMF value at round $t$. Using this AMF regret bound, we can bound the Learner's external regret as:

$$\mathbb{E}\left[R_{\text{ext}}^T\right] = \mathbb{E}\left[\max_{j \in \mathcal{A}} \sum_{t \in [T]} r_{a^t}^t - r_j^t\right] = \mathbb{E}\left[\max_{j \in \mathcal{A}} \sum_{t \in [T]} \ell_j^t(a^t, r^t)\right] \leq 4\sqrt{T \ln |\mathcal{A}|} + \sum_{t \in [T]} w_A^t.$$

It thus remains to show that the AMF value $w_A^t \leq 0$ for all $t$. This holds, since if the Learner knew the Adversary's choice of losses $(r_a^t)_{a \in \mathcal{A}}$ before round $t$, then picking the action $a \in \mathcal{A}$ with the smallest loss $r_a^t$ would get her 0 regret in that round. [4] This gives the in-expectation regret bound; the high-probability bound follows in the same way from Theorem B.2. $\qquad\square$

A bound of $\sqrt{T \ln |\mathcal{A}|}$ is optimal for external regret in the experts learning setting, and so serves to witness the optimality of our framework's general AMF regret bound in Theorem 2.1.

In fact, the above instantiation of Algorithm B.1 yields the classical Exponential Weights algorithm Littlestone and Warmuth [1994]: at each round $t$, the action $a^t$ is sampled with $\Pr[a^t = j] \sim \exp\left(-\eta \sum_{s=1}^{t-1} r_j^s\right)$, for $j \in \mathcal{A}$. We denote this distribution by $\text{EW}_\eta(\pi^{t-1}) \in \Delta(\mathcal{A})$.

Indeed, given the above defined loss $\ell^t$, the Learner solves the following problem at each round:

$$x^t \in \underset{x \in \Delta\mathcal{A}}{\arg\min} \max_{r^t \in [0,1]^{|\mathcal{A}|}} \sum_{j \in \mathcal{A}} \chi_j^t \mathbb{E}_{a \sim x}[r_a^t - r_j^t],$$

where $\chi_j^t = \dfrac{\exp\left(\eta \sum_{s=1}^{t-1}(r_{a^s}^s - r_j^s)\right)}{\sum_{i \in \mathcal{A}} \exp\left(\eta \sum_{s=1}^{t-1}(r_{a^s}^s - r_i^s)\right)} = \dfrac{\exp\left(-\eta \sum_{s=1}^{t-1} r_j^s\right)}{\sum_{i \in \mathcal{A}} \exp\left(-\eta \sum_{s=1}^{t-1} r_i^s\right)}$. That is, the per-coordinate weights $(\chi_j^t)_{j \in \mathcal{A}}$ themselves form the Exponential Weights distribution with rate $\eta$.

For any choice of $r^t$ by the Adversary, the quantity inside the expectation, $\ell_j^t(a, r^t) = r_a^t - r_j^t$, is *antisymmetric* in $a$ and $j$: that is, $\ell_j^t(a, r^t) = -\ell_a^t(j, r^t)$. Due to this antisymmetry, no matter which $r^t$ gets selected by the Adversary, by playing $a \sim \text{EW}_\eta(\pi^{t-1})$ the Learner obtains $\mathbb{E}_{a,j \sim \text{EW}_\eta(\pi^{t-1})}[r_a^t - r_j^t] = 0$, thus achieving the value of the game. It is also easy to see that $x^t = \text{EW}_\eta(\pi^{t-1})$ is the unique choice of $x^t$ that guarantees nonnegative value, hence Algorithm B.1, when specialized to the external regret setting, is *equivalent* to the Exponential Weights Algorithm C.1.

---

[4] Formally, for any vector of actions' losses $r^t$, define $a_{r^t}^* := \arg\min_{a \in \mathcal{A}} r_a^t$, and notice that

$$\min_{a \in \mathcal{A}} \max_{j \in \mathcal{A}} \ell_j^t(a, r^t) \leq \max_{j \in \mathcal{A}} \ell_j^t\left(a_{r^t}^*, r^t\right) = \max_{j \in \mathcal{A}}\left(r_{a_{r^t}^*}^t - r_j^t\right) = \min_{a \in \mathcal{A}} r_a^t - \min_{j \in \mathcal{A}} r_j^t = 0.$$

Hence, the AMF value is indeed nonpositive at each round: $w_A^t = \sup_{r^t \in [0,1]^{|\mathcal{A}|}} \min_{a \in \mathcal{A}} \max_{j \in \mathcal{A}} \ell_j^t(a, r^t) \leq 0$.

# 4 Multicalibration and Multicalibeating

We now apply our framework to derive an online contextual prediction algorithm which simultaneously satisfies a (potentially very large) family of strong adversarial accuracy and calibration conditions. Namely, given an arbitrarily complex family $\mathcal{G}$ of subsets of the context space (we call them "groups", a term from the fairness literature), the predictor will be both calibrated and accurate on each group $g \in \mathcal{G}$ (that is, over those online rounds when the context belongs to $g$).

The accuracy benchmark that we aim to satisfy was recently proposed by Foster and Hart [2021], who called it *calibeating*: given any collection $\mathcal{F}$ of online forecasters, the goal is (intuitively) to "beat" the (squared) error of each $f \in \mathcal{F}$ by at least the *calibration score* of $f$.

In Section 4.1, we use our framework to rederive the online multigroup calibration (known as *multicalibration*) algorithm of Gupta et al. [2022]. In Section 4.2, we show that by appropriately augmenting the original collection of groups $\mathcal{G}$, this algorithm will, *in addition to* multicalibration, calibeat any family of predictors $f \in \mathcal{F}$ on every group $g \in \mathcal{G}$, which we call *multicalibeating*.

## 4.1 Multicalibration

**Setting**   There is a *feature* (or *context*) space $\Theta$ encoding the set of possible feature vectors representing *individuals* $\theta \in \Theta$. There is also a label space $[0,1]$. At every round $t \in [T]$:

1. The Adversary announces a particular individual $\theta^t \in \Theta$, whose label is to be predicted;
2. The Learner predicts a label distribution $x^t$ over $[0,1]$;
3. The Adversary observes $x^t$, and fixes the true label distribution $y^t$ over $[0,1]$;
4. The (pure) guessed label $a^t \sim x^t$ and the (pure) true label $b^t \sim y^t$ are sampled.

**Objective: Multicalibration**   The Learner is initially given an arbitrary collection $\mathcal{G} \subseteq 2^\Theta$ of protected population *groups*. Her goal, *multicalibration*, is empirical calibration not just marginally over the whole population, but also conditionally on individual membership in each $g \in \mathcal{G}$. Formally, for any $n \geq 1$ we let the *n-bucketing* of the label interval $[0,1]$ be its partition into subintervals $[0, 1/n), \ldots, [1 - 2/n, 1 - 1/n), [1 - 1/n, 1]$. The $i^{\text{th}}$ of these intervals (buckets) is denoted $B_n^i$.

**Definition 4.1** (($\alpha, n$)-Multicalibration with respect to $\mathcal{G}$). *Fix a real $\alpha > 0$ and an integer $n \geq 1$. Given the transcript of the interaction $\{(a^t, b^t)\}_{t \in [T]}$, the Learner's sequence of guessed labels $\{a^t\}_{t \in [T]}$ is ($\alpha, n$)-multicalibrated with respect to the collection of groups $\mathcal{G}$ if:*

$$\frac{1}{T}\left|\sum_{t=1}^T \mathbb{1}_{\theta^t \in g} \cdot \mathbb{1}_{a^t \in B_n^i} \cdot (b^t - a^t)\right| \leq \alpha, \text{ for every group } g \in \mathcal{G} \text{ and every bucket } B_n^i \text{ (for } i \in [n]\text{)}.$$

Using our framework, we now derive the guarantee on $\alpha$ that matches that of Gupta et al. [2022].

**Theorem 4.1** (Multicalibration). *Fix a family of groups $\mathcal{G}$, a time horizon $T \geq \ln(2|\mathcal{G}|n)$, and any natural $n, r \geq 1$. Then, our framework's Algorithm B.1 can be instantiated as Algorithm D.1 to produce ($\alpha, n$)-multicalibrated predictions w.r.t. $\mathcal{G}$, where $\alpha$ satisfies (over transcript randomness):*

$$\mathbb{E}[\alpha] \leq \frac{1}{rn} + 4\sqrt{\frac{\ln(2|\mathcal{G}|n)}{T}} \quad \text{and} \quad \Pr\left[\alpha \leq \frac{1}{rn} + 8\sqrt{\frac{1}{T}\ln\left(\frac{2|\mathcal{G}|n}{\delta}\right)}\right] \geq 1 - \delta \,\forall\, \delta \in (0,1).$$

*Proof sketch; see Appendix D for details.* The adversary's strategy space will be $\mathcal{Y} = [0,1]$. The learner will randomize over $\mathcal{A}_r = \{0, 1/(rn), 2/(rn), \ldots, 1\}$ for an integer $r \geq 1$ (to ensure continuity of the loss functions that we are about to define), i.e., her strategy space is $\mathcal{X} = \Delta\mathcal{A}_r$.

*Loss functions:* The definition of multicalibration consists of $2|\mathcal{G}|n$ constraints (one for each $\pm$ sign, group $g$, and bucket $i$) of the following form: $\pm\frac{1}{T}\sum_{t=1}^T \mathbb{1}_{\theta^t \in g} \cdot \mathbb{1}_{a^t \in B_n^i} \cdot (b^t - a^t) \leq \alpha$. Thus, we define (for each $t \in [T]$, $\sigma = \pm 1$, $g$, and $i$) a loss function over $(a^t, b^t) \in \mathcal{A}_r \times \mathcal{Y}$ as: $\ell_{i,g,\sigma}^t(a^t, b^t) := \sigma \cdot \mathbb{1}_{\theta^t \in g} \cdot \mathbb{1}_{a^t \in B_n^i} \cdot (b^t - a^t)$.

Now, defining a $2|\mathcal{G}|n$ dimensional loss vector $\ell^t := \left(\ell_{i,g,\sigma}^t\right)_{i \in [n], g \in \mathcal{G}, \sigma \in \{-1,1\}}$ for each $t \in [T]$ recasts multicalibration in our framework as requiring that $\max_{i \in [n], g \in \mathcal{G}, \sigma \in \{-1,1\}} \sum_{t=1}^T \ell_{i,g,\sigma}^t(a^t, b^t) \leq \alpha T$.

*Bounding the AMF regret:* To bound the Adversary-Moves-First value with these loss functions, suppose the Adversary announces $b^t \in [0, 1]$. Then, we easily see that by (deterministically) responding with $a^t = \operatorname{argmin}_{a \in \mathcal{A}_r} |b^t - a|$, for all $\sigma, g, i$, $\ell^t_{i,g,\sigma}(a^t, b^t) \leq \frac{1}{2rn}$. Hence,

$$w^t_A = \sup_{b^t \in [0,1]} \min_{x^t \in \Delta \mathcal{A}_r} \max_{i \in [n], g \in \mathcal{G}, \sigma \in \{-1,1\}} \mathbb{E}_{a^t \sim x^t} \left[ \ell^t_{i,g,\sigma}\left(a^t, b^t\right) \right] \leq \frac{1}{2rn} \quad \text{for every } t \in [T].$$

Now, for $T \geq \ln(2|\mathcal{G}|n)$, the AMF regret $R^T = \max_{i \in [n], g \in \mathcal{G}, \sigma \in \{-1,1\}} \sum_{t=1}^T \ell^t_{i,g,\sigma}(a^t, b^t) - \sum_{t=1}^T w^t_A$, by our framework's guarantees, satisfies $\mathbb{E}[R^T] \leq 4\sqrt{T \ln(2|\mathcal{G}|n)}$ over the Learner's randomness. Since $\sum_{t=1}^T w^t_A \leq \frac{T}{2rn}$, we get $\mathbb{E}[\max_{i \in [n], g \in \mathcal{G}, \sigma \in \{-1,1\}} \sum_{t=1}^T \ell^t_{i,g,\sigma}(a^t, b^t)] \leq \frac{T}{2rn} + 4\sqrt{T \ln(2|\mathcal{G}|n)}$.

This gives $(\alpha, n)$-multicalibration with $\mathbb{E}[\alpha] \leq \frac{1}{T}\left(\frac{T}{2rn} + 4\sqrt{T \ln(2|\mathcal{G}|n)}\right) = \frac{1}{2rn} + 4\sqrt{\frac{\ln(2|\mathcal{G}|n)}{T}}$. The high-probability bound on $\alpha$ is obtained similarly.

*Simplifying Learner's algorithm:* To attain the AMF value $w^t_A = \frac{1}{2rn}$ at each round, our framework has the Learner solve a linear program (that encodes her minimax strategy). However, she can obtain the *almost* optimal value $\frac{1}{rn}$ *without* solving an LP: this observation gives Algorithm D.1 (see Appendix D). The guarantees on $\alpha$ only differ from optimal ones by replacing $\frac{1}{2rn} \to \frac{1}{rn}$. $\qquad\square$

### 4.2 Multicalibeating

We now give an approach to "beating" arbitrary collections of online forecasters via online multicalibration. The goal, called *calibeating* by Foster and Hart [2021] who introduce the problem, is to make calibrated forecasts that are more accurate than each of an arbitrary set of forecasters, by exactly the calibration error in hindsight of that forecaster. They achieve optimal calibeating bounds for a single forecaster, but their extension to calibeating multiple forecasters incurs at least a polynomial dependence on the number of forecasters. We achieve a logarithmic dependence on the number of forecasters. Additionally, we are able to simultaneously calibeat forecasters on *all* (big enough) subgroups in some set $\mathcal{G}$, with still only a logarithmic dependence on $|\mathcal{G}|$ and the number of forecasters in the group-wise convergence bound. We call this multicalibeating. We now give an overview of our setting, results, and techniques. For full details, see Appendix E.

**Setting** The Learner (predictor $a = \{a^t\}_{t \in [T]}$) and the Adversary (true labels $b = \{b^t\}_{t \in [T]}$) interact in the same way as in Section 4.1, but the Adversary additionally reveals to the Learner a finite set of forecasters $\mathcal{F}$, where each $f \in \mathcal{F}$ is a function $f : \Theta \to D_f$. Here $D_f \subset [0, 1]$ is assumed to be a finite set of all possible forecasts that $f$ makes: it will characterize the *level sets* of $f$. We often suppress the dependence on the transcript, denoting $f^t \in D_f$ the forecast at time $t$.

The Learner's goal is to "improve on" the forecasts of all $f \in \mathcal{F}$, for some suitable scoring of the predictions. We measure the Learner's and the forecasters' accuracy via the squared error, alternatively known as the Brier score.

**Definition 4.2** (Brier Score). *The Brier score of a forecaster $f$ over all rounds $t \in [T]$ is defined as:* $\mathcal{B}^f(\pi^T) := \frac{1}{T} \sum_{t \in [T]} (f^t - b^t)^2$.

The Brier score can be decomposed into so-called *calibration* and *refinement* parts. The former quantifies the extent to which the predictor is calibrated, while the latter expresses the average amount of variance in predictions within every calibration bucket.

To define this decomposition, we need some extra notation. We denote by $S_i$ the subsequence of days on which the Learner's prediction is in bucket $i$.[5] Similarly, $S^d(f)$ (eliding $(f)$ when clear from context) denotes days on which forecaster $f$ predicts $d$. We let $S^d_i(f) = S_i \cap S^d(f)$. Finally, we use bars to indicate average predictions over given subsequences. For instance, $\bar{a}(S)$ is the Learner's average prediction over a given subsequence $S$.

**Definition 4.3** (Calibration and Refinement). *The calibration score $\mathcal{K}$ and refinement score $\mathcal{R}$ of a forecaster $f$ over the full transcript $\pi^T$ are defined as:*

$$\mathcal{K}^f(\pi^T) := \frac{1}{T} \sum_{d \in D_f} |S^d|(d - \bar{b}(S^d))^2, \qquad \mathcal{R}^f(\pi^T) := \frac{1}{T} \sum_{d \in D_f} \sum_{t \in S^d} (b^t - \bar{b}(S^d))^2.$$

---
[5]Note that $S_i$ depends implicitly on the bucketing parameter $n$ and the transcript $\pi^T$.

**Fact 1** (Calibration-Refinement Decomposition of Brier Score [DeGroot and Fienberg, 1983]). $\mathcal{B}^f(\pi^T) = \mathcal{K}^f(\pi^T) + \mathcal{R}^f(\pi^T)$.

The goal of calibeating is to beat the forecaster's Brier score by an amount equal to its calibration score. Or equivalently, to attain a Brier score (almost) equal to the refinement score of the forecaster.

**Definition 4.4** (Calibeating). *The Learner's predictor $a$ is said to $\tau$-calibeat a forecaster $f$ if:* $\mathcal{B}^a(\pi^T) \leq \mathcal{R}^f(\pi^T) + \tau$.

We will now extend the definition of calibeating simultaneously along two natural directions. First, we will want to calibeat multiple forecasters at once. The second extension is that we will want to calibeat the forecasters not just overall, but also on each of the subsequences corresponding to each "population group" $g \in \mathcal{G}$ in a given family of subpopulations $\mathcal{G} \subseteq 2^\Theta$.

**Definition 4.5** (Multicalibeating). *Given a family of forecasters $\mathcal{F}$, groups $\mathcal{G} \subseteq 2^\Theta$, and a mapping $\beta : \mathcal{F} \times \mathcal{G} \to \mathbb{R}_{\geq 0}$, the Learner's predictor $a$ is an $(\mathcal{F}, \mathcal{G}, \beta)$-multicalibeater if for every $g \in \mathcal{G}$:* $\mathcal{B}^a(\pi^T|_{\{t:\theta^t \in g\}}) \leq \min_{f \in \mathcal{F}} \left\{ \mathcal{R}^f(\pi^T|_{\{t:\theta^t \in g\}}) + \beta(f, g) \right\}$

Note that $(\{f\}, \{\Theta\}, \beta(f, \Theta) := \tau)$-multicalibeating is equivalent to $\tau$-calibeating a forecaster $f$.

We first show how to calibeat a single forecaster (Definition 4.4). The modularity of multicalibration will then let us easily extend this result to multiple forecasters and population subgroups.

The idea is to show that if our predictor is multicalibrated with respect to the *level sets* of $f$, then we achieve calibeating. Hébert-Johnson et al. [2018] give a similar bound in the batch setting. We denote the collection of level sets of $f$ as: $\mathcal{S}(f) := \{\theta \in \Theta : f(\theta) = d\}_{d \in D_f}$.

**Theorem 4.2** (Calibeating One Forecaster). *Suppose that the Learner's predictions $a$ are $(\alpha, n)$-multicalibrated on the collection of groups $\mathcal{S}(f) \cup \{\Theta\}$. Then the Learner is $(\alpha, n)$-calibrated on $\Theta$, and she $(\alpha n(|D_f| + 2) + \frac{2}{n})$-calibeats forecaster $f$.*

*Proof sketch.* We show that $a$ has small calibration score, and refinement score close to that of $f$.

*Step 1: Replace $\mathcal{B}^a$ with a surrogate Brier score $\mathcal{B}_n^a$.* Consider a (pseudo-)predictor $\tilde{a}$ given by $\tilde{a}^t = \bar{a}(S_{i_{a^t}})$ for $t \in [T]$ (where $i_{a^t}$ is the bucket of $a^t$). That is, whenever $a^t \in B_n^i$, $\tilde{a}^t$ predicts the average of $a$ over all such rounds $s \in [T]$ that $a^s \in B_n^i$. This is a pseudo-predictor, as the bucket averages of $a$ are unknown until after round $T$. Thus, $\tilde{a}$ has precisely $n$ level sets, unlike $a$. Now, we define $\mathcal{B}_n^a, \mathcal{K}_n^a, \mathcal{R}_n^a$ to be the Brier, calibration, and refinement scores of $\tilde{a}$. We can show $\mathcal{B}^a \leq \mathcal{B}_n^a + 1/n$, allowing us to switch to bounding the more manageable Brier loss $\mathcal{B}_n^a = \mathcal{K}_n^a + \mathcal{R}_n^a$.

*Step 2: Bound the surrogate calibration score $\mathcal{K}_n^a$.* Since the Learner is $(\alpha, n)$-calibrated on the domain $\Theta$, the calibration error per level set is at most $\alpha$. There are $n$ level sets, so $\mathcal{K}_n^a \leq \alpha n$.

*Step 3: Bound the surrogate refinement score $\mathcal{R}_n^a$.* We connect $\mathcal{R}^f$ and $\mathcal{R}_n^a$ via a *joint* refinement score: $\mathcal{R}^{f \times a}$, which measures the average variance of the partition generated by all intersections of the level sets of $a$ and $f$. The finer the partition, the smaller the refinement score, so $\mathcal{R}^f \geq \mathcal{R}^{f \times a}$. Next, informally, multicalibration ensures that $a$ has already "captured" most of the variance explained by $f$. Therefore, refining $a$'s level sets by $f$ does little to reduce variance. More precisely, we show that $\mathcal{R}_n^a \leq \mathcal{R}^{f \times a} + \alpha n(|D_f| + 1) + \frac{1}{n}$. Combining with our previous inequality, we have: $\mathcal{R}_n^a \leq \mathcal{R}^f + \alpha n(|D_f| + 1) + \frac{1}{n}$.

Combining the above, we get: $\mathcal{B}^a \leq \mathcal{R}_n^a + \mathcal{K}_n^a + \frac{1}{n} \leq (\mathcal{R}^f + \alpha n(|D_f| + 1) + \frac{1}{n}) + \alpha n + \frac{1}{n}$. $\quad\square$

**Calibeating many forecasters** Generalizing the above construction, we can easily calibeat any collection of forecasters $\mathcal{F}$ on the entire context space $\Theta$: it suffices to ask for multicalibration with respect to the level sets of *all* forecasters, i.e. $\left( \bigcup_{f \in \mathcal{F}} \mathcal{S}(f) \right) \cup \{\Theta\}$. Theorem 4.2 applies separately to each $f$; the only degradation in the guarantees will come in the form of a larger $\alpha$, since we are asking for multicalibration with respect to more groups than before. But this effect will be small, since $\alpha$ depends on the number of required groups $|\mathcal{G}'|$ as $O(\sqrt{\ln |\mathcal{G}'|})$. See Corollary E.2.

However, to fully satisfy Definition 4.5 of multicalibeating, we need to calibeat all $f \in \mathcal{F}$ on all groups $g \in \mathcal{G}$ in a given collection $\mathcal{G} \subseteq 2^\Theta$. For that, we simply extend the above construction by requiring multicalibration with respect to all pairwise *intersections* of the forecasters' level sets with

the groups $g \in \mathcal{G}$. By further augmenting this collection with the protected groups $\mathcal{G}$ themselves, we finally achieve our ultimate goal: *simultaneous* multicalibeating and multicalibration.

**Theorem 4.3** (Multicalibeating + Multicalibration). *Let $\mathcal{G} \subseteq 2^{\Theta}$, and $\mathcal{F}$ some set of forecasters $f : \Theta \to D_f$. The multicalibration algorithm on $\mathcal{G}' := \left( \bigcup_{f \in \mathcal{F}} \{g \cap S : (g, S) \in \mathcal{G} \times \mathcal{S}(f)\} \right) \cup \mathcal{G}$ with parameters $r, n \geq 1$, after $T$ rounds, attains expected $(\mathcal{F}, \mathcal{G}, \beta)$-multicalibeating, where:* [6]*

$$\mathbb{E}[\beta(f,g)] \leq \tfrac{2}{n} + \tfrac{|D_f|+2}{r \cdot |S(g)|/T} + 4n(|D_f|+2)\sqrt{\tfrac{1}{|S(g)|^2/T} \ln \left( 2n|\mathcal{G}|(1 + \sum_f |D_f|) \right)} \; \forall \, f \in \mathcal{F}, g \in \mathcal{G},$$

*while maintaining $(\alpha, n)$-multicalibration on $\mathcal{G}$, with:* $\mathbb{E}[\alpha] \leq \tfrac{1}{rn} + 4\sqrt{\tfrac{1}{T} \ln \left( 2n|\mathcal{G}|(1 + \sum_f |D_f|) \right)}$.

In particular, for any group $g$ occurring more than $T^{-1/2}$ of the time, we asymptotically converge to $\frac{1}{n}$-calibeating as $T \to \infty$, thus combining the goals of online multicalibration and multigroup regret.

# 5 Polytope Blackwell Approachability

Consider a setting where the Learner and the Adversary are playing a repeated game with *vector-valued* payoffs, in which the Learner always goes first and aims to force the average payoff over the entire interaction to approach a given convex set. Blackwell's Theorem [1956] states that a convex set is approachable if and only if it is *response-satisfiable* (roughly, for any choice of the Adversary, the Learner has a response forcing the one-round payoff inside the convex set). The rate of approachability typically depends on the dimension of the payoff vectors.

This is a specialization of our framework to the case of a fixed environment across rounds. Thus our framework can be used to obtain a *dimension-independent* rate bound in the fundamental case where the approachable set is a convex *polytope*. Our bound is only logarithmic in the polytope's number of facets, and is achievable via an efficient convex-programming based algorithm.

Let us formalize our setting. In rounds $t = 1, 2, \ldots$, the Learner and the Adversary play a repeated game. Their respective pure strategy sets are $\mathcal{A}$ and $\mathcal{Y}$, where $\mathcal{A}$ is a finite set and $\mathcal{Y} \subseteq \mathbb{R}^m$ (for some integer $m \geq 1$) is convex and compact. The game's utility function is $\lambda$-dimensional (for some integer $\lambda \geq 1$), continuous, concave in the second argument, and is denoted by $u : \mathcal{A} \times \mathcal{Y} \to \mathbb{R}^\lambda$. At each round $t$, the Learner plays a mixed strategy $x^t \in \Delta\mathcal{A}$, the Adversary responds with some $y^t \in \mathcal{Y}$, and the Learner then samples a pure action $a^t \sim x^t$. This gives rise to the utility vector $u(a^t, y^t)$. The *average play* up to any round $t \geq 1$ is then defined as $\bar{u}^t = \frac{1}{t} \sum_{s=1}^{t} u(a^s, y^s)$.

The target convex set that the Learner wants to approach is a polytope $\mathcal{P}(\mathcal{H}) \subseteq \mathbb{R}^\lambda$, defined as the intersection of a finite collection of halfspaces $\mathcal{H} = (h_{\alpha,\beta})$, where for any given $\alpha \in \mathbb{R}^\lambda, \beta \in \mathbb{R}$ we denote $h_{\alpha,\beta} = \{x \in \mathbb{R}^\lambda : \langle \alpha, x \rangle - \beta \leq 0\}$. Finally, by way of normalization, consider any two dual norms $|| \cdot ||_p$ and $|| \cdot ||_q$. We require, first, that $||\alpha||_p \leq 1$ and $|\beta| \leq 1$ for each halfspace $h_{\alpha,\beta} \in \mathcal{H}$; and second, that the payoffs be in the $|| \cdot ||_q$-unit ball: $||u(a, y)||_q \leq 1$ for $a \in \mathcal{A}, y \in \mathcal{Y}$.

**Theorem 5.1** (Polytope Blackwell Approachability). *Suppose the target convex polytope $\mathcal{P}(\mathcal{H})$ is* response-satisfiable, *in the sense that for any Adversary's action $y \in \mathcal{Y}$, the Learner has a mixed response $x \in \Delta\mathcal{A}$ that places the expected payoff inside $\mathcal{P}(\mathcal{H})$: that is, $\mathbb{E}_{a \sim x}[u(a, y)] \in \mathcal{P}(\mathcal{H})$.*

*Then, $\mathcal{P}(\mathcal{H})$ is* approachable, *both in expectation and with high probability with respect to the transcript of the interaction. Namely, the Learner has an efficient convex programming based algorithm which simultaneously offers both following guarantees (see Appendix F for the proof):*

1. *For any margin $\epsilon > 0$, the average play $\bar{u}^t$ up to any round $t \geq \frac{64 \ln |\mathcal{H}|}{\epsilon^2}$ will satisfy $\mathbb{E} \left[ \max_{h_{\alpha,\beta} \in \mathcal{H}} (\langle \alpha, \bar{u}^t \rangle - \beta) \right] \leq \epsilon$.*

2. *For any $\delta \in (0, 1)$, the average play $\bar{u}^t$ up to any round $t \geq \ln |\mathcal{H}|$ will satisfy $\max_{h_{\alpha,\beta} \in \mathcal{H}} (\langle \alpha, \bar{u}^t \rangle - \beta) \leq 16 \sqrt{\frac{1}{T} \ln \left( \frac{|\mathcal{H}|}{\delta} \right)}$ with probability at least $1 - \delta$.*

---

[6] $S(g)$ denotes the subsequence of days on which a group $g$ occurs, suppressing dependence on transcript.

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
