# Online Minimax Multiobjective Optimization: Multicalibeating and Other Applications — Supplementary Material

**Daniel Lee**[1]**, Georgy Noarov**[1]**, Mallesh Pai**[2]**, Aaron Roth**[1]
[1] University of Pennsylvania, [2] Rice University
daniellee@alumni.upenn.edu, gnoarov@seas.upenn.edu,
mallesh.pai@rice.edu, aaroth@cis.upenn.edu

## A   Additional Related Work

Papers by Azar et al. [2014] and Kesselheim and Singla [2020] study a related problem: an online setting with vector-valued losses, where the goal is to minimize the $\ell_\infty$ norm of the accumulated loss vector (they also consider other $\ell_p$-norms). However, they study an incomparable benchmark that in our notation would be written as $\min_{x^* \in \mathcal{X}} \max_{j \in [d]} \frac{1}{T} \sum_{t=1}^{T} \ell_j(x^*, y^t)$ (which is well-defined in their setting, where loss functions $\ell^t = \ell$ and action sets $\mathcal{X}^t = \mathcal{X}, \mathcal{Y}^t = \mathcal{Y}$ are fixed throughout the interaction). On the one hand, this benchmark is stronger than ours in the sense that the maximum over coordinates is taken outside the sum over time, whereas our benchmark considers a "greedy" per-round maximum. On the other hand, in our setting the game can be different at every round, so our benchmark allows a comparison to a different action at each round rather than a single fixed action. In the setting of Kesselheim and Singla [2020], it is impossible to give any regret bound to their benchmark, so they derive an algorithm obtaining a $\log(d)$ *competitive ratio* to this benchmark. In contrast, our benchmark admits a regret bound. Hence, our results are quite different in kind despite the outward similarity of the settings: none of our applications follow from their theorems (since in all of our applications, we derive regret bounds).

A different line of work [Rakhlin et al., 2010, 2011] takes a very general minimax approach towards deriving bounds in online learning, including regret minimization, calibration, and approachability. Their approach is substantially more powerful than the framework we introduce here (e.g. it can be used to derive bounds for infinite dimensional problems, and characterizes online learnability in the sense that it can also be used to prove lower bounds). However, it is also correspondingly more complex, and requires analyzing the continuation value of a $T$ round dynamic program. Such analyses are generally technically challenging; as an example, a recent line of work by Drenska and Kohn [2020] and Kobzar et al. [2020] considers a Rakhlin et al.-style minimax formulation of the standard experts problem, and shows how to find nonlinear PDE-based minimax solutions for the Learner and the Adversary that can be optimal not just asymptotically in the number of experts (dimensions) $d$, but also nonasymptotically for small $d$ such as 2 or 3; their PDE approach is also conducive to bounding not just the maximum regret across dimensions, but also more general functions of the individual dimensions' losses.

Overall, results derived from the Rakhlin et al. framework (with some notable exceptions, including Rakhlin et al. [2012]) are generically nonconstructive, whereas our framework is simple and inherently constructive, in that the algorithm derives from repeatedly solving a one-round stage zero-sum game. Relative to this literature, we view our framework as a "user-friendly" power tool that can be used to derive a wide variety of algorithms and bounds without much additional work — at the cost of not being universally expressive.

36th Conference on Neural Information Processing Systems (NeurIPS 2022).

# B  The General Framework with Extensions to Probabilistic and Approximate Learners: Full Proofs and Algorithms

## B.1  Omitted Proofs from Section 2

*Proof of Lemma 2.1.* After taking the log and dividing by $\eta$, this lemma follows from the following chain:

$$\exp\left(\eta R^T\right) = \exp\left(\eta \max_{j\in[d]} R_j^T\right) = \exp\left(\max_{j\in[d]} \eta R_j^T\right) = \max_{j\in[d]} \exp\left(\eta R_j^T\right) \le \sum_{j\in[d]} \exp\left(\eta R_j^T\right) = L^T.$$

$\square$

*Proof of Lemma 2.2.* By definition of the surrogate loss, we have:

$$\begin{aligned}
L^t - L^{t-1} &= \sum_{j\in[d]} \exp\left(\eta R_j^t\right) - \sum_{j\in[d]} \exp\left(\eta R_j^{t-1}\right), \\
&= \sum_{j\in[d]} \exp\left(\eta R_j^{t-1} + \eta\left(\ell_j^t\left(x^t, y^t\right) - w_A^t\right)\right) - \sum_{j\in[d]} \exp\left(\eta R_j^{t-1}\right), \\
&= \sum_{j\in[d]} \exp\left(\eta R_j^{t-1}\right)\left(\exp\left(\eta\left(\ell_j^t\left(x^t, y^t\right) - w_A^t\right)\right) - 1\right).
\end{aligned}$$

Using the fact that $\exp(x) - 1 \le x + x^2$ for $|x| \le 1$, we have, for $\eta \cdot 2C \le 1$,

$$\begin{aligned}
&\le \sum_{j\in[d]} \exp\left(\eta R_j^{t-1}\right)\left(\eta\left(\ell_j^t(x^t, y^t) - w_A^t\right) + \eta^2\left(\ell_j^t(x^t, y^t) - w_A^t\right)^2\right), \\
&\le \eta \sum_{j\in[d]} \exp\left(\eta R_j^{t-1}\right)\left(\ell_j^t\left(x^t, y^t\right) - w_A^t\right) + \eta^2(2C)^2 L^{t-1}.
\end{aligned}$$

$\square$

*Proof of Lemma 2.3.* We begin by recalling that $L^0 = d$. Thus, the desired bound on $L^T$ follows via Lemma 2.2 and a telescoping argument, if only we can show that for every $t \in [T]$ the Learner has an action $x^t \in \mathcal{X}^t$ which guarantees that for any $y^t \in \mathcal{Y}^t$,

$$\eta \sum_{j\in[d]} \exp\left(\eta R_j^{t-1}\right)\left(\ell_j^t(x^t, y^t) - w_A^t\right) \le 0.$$

To this end, we define a zero-sum game between the Learner and the Adversary, with action space $\mathcal{X}^t$ for the Learner and $\mathcal{Y}^t$ for the Adversary, and with the objective function (which the Adversary wants to maximize and the Learner wants to minimize):

$$u^t(x, y) := \sum_{j\in[d]} \exp\left(\eta R_j^{t-1}\right)\left(\ell_j^t(x, y) - w_A^t\right), \text{ for all } x \in \mathcal{X}^t, y \in \mathcal{Y}^t.$$

Recall from the definition of our framework that $\mathcal{X}^t, \mathcal{Y}^t$ are convex, compact and finite-dimensional, as well as that each $\ell_j^t$ is continuous, convex in the first argument, and concave in the second argument. Since $u^t$ is defined as an affine function of the individual coordinate functions $\ell_j^t$, $u^t$ is also convex-concave and continuous. This means that we may invoke Sion's Minimax Theorem:

**Fact 1** (Sion's Minimax Theorem). *Given finite-dimensional convex compact sets $\mathcal{X}, \mathcal{Y}$, and a continuous function $f : \mathcal{X} \times \mathcal{Y} \to \mathbb{R}$ which is convex in the first argument and concave in the second argument, it holds that*

$$\min_{x\in\mathcal{X}} \max_{y\in\mathcal{Y}} f(x, y) = \max_{y\in\mathcal{Y}} \min_{x\in\mathcal{X}} f(x, y).$$

Using Sion's Theorem to switch the order of play (so that the Adversary is compelled to move first), and then recalling the definition of $w_A^t$ (the value of the maximum coordinate value of $\ell^t$ that the Learner can obtain when the Adversary is compelled to move first), we obtain:[B.1]

$$\min_{x^t \in \mathcal{X}^t} \max_{y^t \in \mathcal{Y}^t} u^t\left(x^t, y^t\right) = \max_{y^t \in \mathcal{Y}^t} \min_{x^t \in \mathcal{X}^t} u^t\left(x^t, y^t\right)$$

$$= \max_{y^t \in \mathcal{Y}^t} \min_{x^t \in \mathcal{X}^t} \sum_{j' \in [d]} \exp\left(\eta R_{j'}^{t-1}\right) \cdot \left(\ell_{j'}^t\left(x^t, y^t\right) - w_A^t\right),$$

$$\leq \sup_{y^t \in \mathcal{Y}^t} \min_{x^t \in \mathcal{X}^t} \sum_{j' \in [d]} \exp\left(\eta R_{j'}^{t-1}\right) \cdot \max_{j \in [d]}\left(\ell_j^t\left(x^t, y^t\right) - w_A^t\right),$$

$$= \sum_{j' \in [d]} \exp\left(\eta R_{j'}^{t-1}\right) \cdot \sup_{y^t \in \mathcal{Y}^t} \min_{x^t \in \mathcal{X}^t} \max_{j \in [d]}\left(\ell_j^t\left(x^t, y^t\right) - w_A^t\right),$$

$$= \sum_{j' \in [d]} \exp\left(\eta R_{j'}^{t-1}\right) \cdot \left(w_A^t - w_A^t\right),$$

$$= 0.$$

Thus, the Learner can ensure that $L^t \leq \left(4\eta^2 C^2 + 1\right) L^{t-1}$ by playing at every round $t$:

$$x^t \in \operatorname*{argmin}_{x \in \mathcal{X}^t} \max_{y \in \mathcal{Y}^t} u^t(x, y).$$

This concludes the proof. $\qquad\square$

**An equivalent description of Learner's space of minimax optimal strategies at each round** $t$
We observe that the Learner's optimal action at each round, derived in the proof, can be expressed without any reference to the quantities $w_A^t$:

$$
\begin{aligned}
x^t \ \in\ & \operatorname*{argmin}_{x \in \mathcal{X}^t} \max_{y \in \mathcal{Y}^t} \sum_{j \in [d]} \exp(\eta R_j^{t-1})(\ell_j^t(x, y) - w_A^t), \\
=\ & \operatorname*{argmin}_{x \in \mathcal{X}^t} \max_{y \in \mathcal{Y}^t} \sum_{j \in [d]} \exp(\eta R_j^{t-1})\ell_j^t(x, y), \\
=\ & \operatorname*{argmin}_{x \in \mathcal{X}^t} \max_{y \in \mathcal{Y}^t} \sum_{j \in [d]} \frac{\exp\left(\eta \sum_{s=1}^{t-1} \ell_j^s(x^s, y^s)\right) \ell_j^t(x, y)}{\exp\left(\eta \sum_{s=1}^{t-1} w_A^s\right)}, \\
=\ & \operatorname*{argmin}_{x \in \mathcal{X}^t} \max_{y \in \mathcal{Y}^t} \sum_{j \in [d]} \exp\left(\eta \sum_{s=1}^{t-1} \ell_j^s(x^s, y^s)\right) \ell_j^t(x, y), \\
=\ & \operatorname*{argmin}_{x \in \mathcal{X}^t} \max_{y \in \mathcal{Y}^t} \sum_{j \in [d]} \frac{\exp\left(\eta \sum_{s=1}^{t-1} \ell_j^s(x^s, y^s)\right)}{\sum_{i \in [d]} \exp\left(\eta \sum_{s=1}^{t-1} \ell_i^s(x^s, y^s)\right)} \ell_j^t(x, y).
\end{aligned}
$$

The weights placed on the loss coordinates $\ell_j^s(x^t, y^t)$ in the final expression form a probability distribution which should remind the reader of the well known Exponential Weights distribution.

## B.2 Extensions

Before presenting applications of our framework, we pause to discuss two natural extensions that are called for in some of our applications. Both extensions only require very minimal changes to the notation in Section 2.1 and to the general algorithmic framework in Section 2.2.

We begin by discussing, in Section B.2.1, how to adapt our framework to the setting where the Learner is allowed to randomize at each round amongst a finite set of actions, and wishes to obtain

---

[B.1]Note that in the third step, $\max_{y^t \in \mathcal{Y}^t}$ turns into $\sup_{y^t \in \mathcal{Y}^t}$. This is because after each $\left(\ell_{j'}^t\left(x^t, y^t\right) - w_A^t\right)$ is replaced with $\max_j\left(\ell_j^t\left(x^t, y^t\right) - w_A^t\right)$, the maximum over $y$ generally becomes unachievable (recall Footnote 1).

probabilistic guarantees for her AMF regret with respect to her randomness. This will be useful in all three of our applications.

We then proceed to show, in Section B.2.2, that our AMF regret bounds are robust to the case in which at each round, the Learner, who is playing according to the general Algorithm 1 given above, computes and plays according to an approximate (rather than exact) minimax strategy. This is useful for settings where it may be desirable (for computational or other reasons) to implement our algorithmic framework approximately, rather than exactly. In particular, in one of our applications — mean multicalibration, which is discussed in Section 4.1 — we will illustrate this point by deriving a multicalibration algorithm that has the Learner play only extremely (computationally and structurally) simple strategies, at the cost of adding an arbitrarily small term to the multicalibration bounds, compared to the Learner that plays the exact minimax equilibrium.

### B.2.1 Performance Bounds for a Probabilistic Learner

So far, we have described the interaction between the Learner and the Adversary as deterministic. In many applications, however, the convex action space for the Learner is the simplex over some finite set of base actions, representing *probability distributions* over actions. In this case, the Adversary chooses his action in response to the *probability distribution* over base actions chosen by the Learner, at which point the Learner samples a single base action from her chosen distribution.

We will use the following notation. The Learner's pure action set at time $t$ is denoted by $\mathcal{A}^t$. Before each round $t$, the Adversary reveals a vector valued loss function $\ell^t : \mathcal{A}^t \times \mathcal{Y}^t \to [-C, C]^d$. At the beginning of round $t$, the Learner chooses a probabilistic mixture over her action set $\mathcal{A}^t$, which we will usually denote as $x^t \in \Delta\mathcal{A}^t$; after the Adversary has made his move, the Learner samples her pure action $a^t$ for the round, which is recorded into the transcript of the interaction.

The redefined vector valued losses $\ell^t$ now take as their first argument a *pure action* $a \in \mathcal{A}^t$. We extend this to $\mathcal{X}^t := \Delta\mathcal{A}^t$ as $\ell^t(x^t, y^t) := \mathbb{E}_{a^t \sim x^t}[\ell^t(a^t, y^t)]$ for any $x^t \in \Delta\mathcal{A}^t$. In this notation, holding the second argument fixed, the loss function is linear (hence convex and continuous) and has a convex, compact domain (the simplex $\Delta\mathcal{A}^t$). Using this extended notation, it is now easy to see how to define the probabilistic analog of the AMF value.

**Definition B.1** (Probabilistic AMF Value)**.**

$$w_A^t := \sup_{y^t \in \mathcal{Y}^t} \min_{x^t \in \mathcal{X}^t} \max_{j \in [d]} \ell_j^t(x^t, y^t) = \sup_{y^t \in \mathcal{Y}^t} \min_{x^t \in \Delta\mathcal{A}^t} \max_{j \in [d]} \mathbb{E}_{a^t \sim x^t} \left[ \ell_j^t(a^t, y^t) \right].$$

For a more detailed discussion of the probabilistic setting, please refer to Appendix B.3.

**Adapting the algorithm to the probabilistic Learner setting**   Above, Algorithm 1 was given for the deterministic case of our framework. In the probabilistic setting, when computing the probability distribution for the current round, the Learner should take into account the *realized* losses from the past rounds. We present the modified algorithm below.

---

**Algorithm B.1:** General Algorithm for the *Probabilistic* Learner

---

**for** rounds $t = 1, \ldots, T$ **do**

Learn adversarially chosen $\mathcal{A}^t, \mathcal{Y}^t$, and vector loss function $\ell^t(\cdot, \cdot) : \mathcal{A}^t \times \mathcal{Y}^t \to [-C, C]^d$.

Let

$$\chi_j^t := \frac{\exp\left(\eta \sum_{s=1}^{t-1} \ell_j^s(a^s, y^s)\right)}{\sum_{i \in [d]} \exp\left(\eta \sum_{s=1}^{t-1} \ell_i^s(a^s, y^s)\right)} \text{ for } j \in [d].$$

Select a mixed action $x^t \in \Delta\mathcal{A}^t$, where

$$x^t \in \operatorname*{argmin}_{x \in \Delta\mathcal{A}^t} \max_{y \in \mathcal{Y}^t} \sum_{j \in [d]} \chi_j^t \cdot \ell_j^t(x, y).$$

Observe the Adversary's selection of $y^t \in \mathcal{Y}^t$.

Sample pure action $a^t \sim x^t$.

---

**Probabilistic performance guarantees** Algorithm B.1 provides two crucial blackbox guarantees to the probabilistic Learner. First, the guarantees on Algorithm 1 from Theorem 2.1 almost immediately translate into a bound on the *expected* AMF regret of the Learner who uses Algorithm B.1, over the randomness in her actions. Second, a *high-probability* AMF regret bound, also over the Learner's randomness, can be derived in a straightforward way.

**Theorem B.1** (In-Expectation Bound). *Given* $T \geq \ln d$, *Algorithm B.1 with learning rate* $\eta = \sqrt{\frac{\ln d}{4TC^2}}$ *guarantees that ex-ante, with respect to the randomness in the Learner's realized outcomes, the expected AMF regret is bounded as:*

$$\mathbb{E}\left[R^T\right] \leq 4C\sqrt{T \ln d}.$$

*Proof Sketch.* Using Jensen's inequality to switch expectations and exponentials, it is easy to modify the proof of Lemma 2.1 to obtain the following in-expectation bound:

$$\mathbb{E}\left[R^T\right] \leq \frac{\ln \mathbb{E}\left[L^T\right]}{\eta}.$$

The rest of the proof is similar to the proofs of Lemma 2.2 and Lemma 2.3. □

**Theorem B.2** (High-Probability Bound). *Fix any* $\delta \in (0,1)$. *Given* $T \geq \ln d$, *Algorithm B.1 with learning rate* $\eta = \sqrt{\frac{\ln d}{4TC^2}}$ *guarantees that the AMF regret will satisfy, with ex-ante probability* $1 - \delta$ *over the randomness in the Learner's realized outcomes,*

$$R^T \leq 8C\sqrt{T \ln\left(\frac{d}{\delta}\right)}.$$

*Proof Sketch.* The proof proceeds by constructing a martingale with bounded increments that tracks the increase in the surrogate loss $L^T$, and then using Azuma's inequality to conclude that the final surrogate loss (and hence the AMF regret) is bounded above with high probability. For a detailed proof, see Appendix B.3. □

### B.2.2 Performance Bounds for a Suboptimal Learner

Our general Algorithms 1 and B.1 involve the Learner solving a convex program at each round in order to identify her minimax optimal strategy. However, in some applications of our framework it may be necessary or desirable for the Learner to restrict herself to playing *approximately* minimax optimal strategies instead of exactly optimal ones. This can happen for a variety of reasons:

1. *Computational efficiency.* While the convex program that the Learner must solve at each round is polynomial-sized in the description of the environment, one may wish for a better running time dependence — e.g. in settings in which the action space for the Learner is exponential in some other relevant parameter of the problem. In such cases, we will want to trade off run-time for approximation error in the minimax equilibrium computation at each round.

2. *Structural simplicity of strategies.* One may wish to restrict the Learner to only playing "simple" strategies (for example, distributions over actions with small support), or more generally, strategies belonging to a certain predefined strict subset of the Learner's strategy space. This subset may only contain approximately optimal minimax strategies.

3. *Numerical precision.* As the convex programs solved by the Learner at each round generally have irrational coefficients (due to the exponents), using finite-precision arithmetic to solve these programs will lead to a corresponding precision error in the solution, making the computed strategy only approximately minimax optimal for the Learner. This kind of approximation error can generally be driven to be arbitrarily small, but still necessitates being able to reason about approximate solutions.

Given a suboptimal instantiation of Algorithm 1 or B.1, we thus want to know: how much worse will its achieved regret bound be, compared to the existential guarantee? We will now address this

question for both the deterministic setting of Sections 2.1 and 2.2, and the probabilistic setting of Section B.2.1.

Recall that at each round $t \in [T]$, both Algorithm 1 and Algorithm B.1 (with the weights $\chi_j^t$ defined accordingly) have the Learner solve for the minimizer $x$ of the function $\psi^t : \mathcal{X}^t \to [-C, C]$ defined as:

$$\psi^t(x) := \max_{y \in \mathcal{Y}^t} \sum_{j \in [d]} \chi_j^t \cdot \ell_j^t(x, y).$$

The range of $\psi^t$ is $[-C, C]$ as indicated, since it is a linear combination of loss coordinates $\ell_j^t(x, y) \in [-C, C]$, where the weights $(\chi_1^t, \dots, \chi_d^t)$ form a probability distribution over $[d]$.

Now suppose the Learner ends up playing actions $x^1, \dots, x^T$ which do not necessarily minimize the respective objectives $\psi^t(\cdot)$. The following definition helps capture the degree of suboptimality in the Learner's play at each round.

**Definition B.2** (Achieved AMF Value Bound). *Consider any round $t \in [T]$, and suppose the Learner plays action $x^t \in \mathcal{X}^t$ at round $t$. Then, any number*

$$w_{\mathrm{bd}}^t \in \left[ \psi^t(x^t), C \right]$$

*is called an* achieved AMF value bound *for round $t$.*

This definition has two aspects. Most importantly, $w_{\mathrm{bd}}^t$ upper bounds the Learner's *achieved* objective function value at round $t$. Furthermore, we restrict $w_{\mathrm{bd}}^t$ to be $\leq C$ — otherwise it would be a meaningless bound as the Learner gets objective value $\leq C$ no matter what $x^t$ she plays.

We now formulate the desired bounds on the performance of a suboptimal Learner. The upshot is that for a suboptimal Learner, the bounds of Theorems 2.1, B.1, B.2 hold with each $w_A^t$ replaced with the corresponding achieved AMF bound $w_{\mathrm{bd}}^t$.

**Theorem B.3** (Bounds for a Suboptimal Learner). *Consider a Learner who does not necessarily play optimally at all rounds, and a sequence $w_{\mathrm{bd}}^1, \dots, w_{\mathrm{bd}}^T$ of achieved AMF value bounds.*

*In the deterministic setting, the Learner achieves the following regret bound analogous to Theorem 2.1:*

$$\max_{j \in [d]} \sum_{t=1}^{T} \ell_j^t(x^t, y^t) \leq \sum_{t=1}^{T} w_{\mathrm{bd}}^t + 4C\sqrt{T \ln d}.$$

*In the probabilistic setting, the Learner achieves the following in-expectation regret bound analogous to Theorem B.1:*

$$\mathbb{E}\left[ \max_{j \in [d]} \sum_{t=1}^{T} \ell_j^t(a^t, y^t) \right] \leq \sum_{t=1}^{T} w_{\mathrm{bd}}^t + 4C\sqrt{T \ln d},$$

*and the following high-probability bound analogous to Theorem B.2:*

$$\max_{j \in [d]} \sum_{t=1}^{T} \ell_j^t(a^t, y^t) \leq \sum_{t=1}^{T} w_{\mathrm{bd}}^t + 8C\sqrt{T \ln\left(\frac{d}{\delta}\right)} \text{ with probability } \geq 1 - \delta, \text{ for any } \delta \in (0, 1).$$

*Proof Sketch.* We use the deterministic case for illustration. The main idea is to redefine the Learner's regret to be relative to her achieved AMF value bounds $(w_{\mathrm{bd}}^t)_{t \in [T]}$ rather than the AMF values $(w_A^t)_{t \in [T]}$. Namely, we let $R_{\mathrm{bd}}^t := \max_{j \in [d]} (R_{\mathrm{bd}}^t)_j$, where $(R_{\mathrm{bd}}^t)_j :=$ $\sum_{s=1}^{t} \ell_j^s(x^s, y^s) - \sum_{s=1}^{t} w_{\mathrm{bd}}^s$. The surrogate loss is defined in the same way as before, namely $L_{\mathrm{bd}}^t := \sum_{j \in [d]} \exp\left( \eta \cdot (R_{\mathrm{bd}}^t)_j \right)$.

First, Lemma 2.1 still holds: $R_{\mathrm{bd}}^T \leq \left( \ln L_{\mathrm{bd}}^T \right) / \eta$, with the same proof. Lemma 2.2 also holds after replacing each $w_A^t$ with $w_{\mathrm{bd}}^t$: namely, $L_{\mathrm{bd}}^t \leq \left( 4\eta^2 C^2 + 1 \right) L_{\mathrm{bd}}^{t-1} + \eta \sum_{j \in [d]} \exp\left( \eta \left( R_{\mathrm{bd}}^{t-1} \right)_j \right) \cdot$ $\left( \ell_j^t(x^t, y^t) - w_{\mathrm{bd}}^t \right)$. The proof is almost the same: we formerly used $w_A^t \leq C$, and now use that $w_{\mathrm{bd}}^t \leq C$ by Definition B.2.

Now, following the proofs of Lemma 2.3 and Theorem 2.1, to obtain the declared regret bound it suffices to show for $t \in [T]$ that the Learner's action $x^t$ guarantees $\sum_{j \in [d]} \exp\left(\eta \left(R_{\mathrm{bd}}^{t-1}\right)_j\right) \cdot \left(\ell_j^t\left(x^t, y^t\right) - w_{\mathrm{bd}}^t\right) \le 0$, no matter what $y^t$ is played by the Adversary. For any $y^t \in \mathcal{Y}^t$, we can rewrite this objective as:

$$\sum_{j \in [d]} \exp\left(\eta \left(R_{\mathrm{bd}}^t\right)_j\right) \cdot \left(\ell_j^t\left(x^t, y^t\right) - w_{\mathrm{bd}}^t\right) = \frac{\sum_{i \in [d]} \exp\left(\eta \sum_{s=1}^{t-1} \ell_i^s(x^s, y^s)\right)}{\exp\left(\sum_{s=1}^{t-1} w_{\mathrm{bd}}^s\right)} \sum_{j \in [d]} \chi_j^t \cdot \left(\ell_j^t(x^t, y^t) - w_{\mathrm{bd}}^t\right).$$

It now follows that action $x^t$ achieves $\sum_{j \in [d]} \exp\left(\eta \left(R_{\mathrm{bd}}^{t-1}\right)_j\right) \cdot \left(\ell_j^t\left(x^t, y^t\right) - w_{\mathrm{bd}}^t\right) \le 0$, from observing that:

$$\sum_{j \in [d]} \chi_j^t \cdot \left(\ell_j^t(x^t, y^t) - w_{\mathrm{bd}}^t\right) = \sum_{j \in [d]} \chi_j^t \cdot \ell_j^t(x^t, y^t) - w_{\mathrm{bd}}^t \le \psi^t(x^t) - w_{\mathrm{bd}}^t \le 0,$$

where the final inequality holds since the Learner achieves AMF value bound $w_{\mathrm{bd}}^t$ at round $t$. $\qquad\square$

## B.3 Omitted Proofs and Details from Section B.2.1: Bounds for the Probabilistic Learner

First, we define our probabilistic setting, emphasizing the differences to the deterministic protocol. At each round $t \in [T]$, the interaction between the Learner and the Adversary proceeds as follows:

1. At the beginning of each round $t$, the Adversary selects an environment consisting of the following, and reveals it to the Learner:

   (a) The Learner's *simplex action set* $\mathcal{X}^t = \Delta\mathcal{A}^t$, where $\mathcal{A}^t$ is a finite set of pure actions;

   (b) The Adversary's convex compact action set $\mathcal{Y}^t$, embedded in a finite-dimensional Euclidean space;

   (c) A vector valued loss function $\ell^t(\cdot, \cdot) : \mathcal{A}^t \times \mathcal{Y}^t \to [-C, C]^d$. Every dimension $\ell_j^t(\cdot, \cdot) : \mathcal{A}^t \times \mathcal{Y}^t \to [-C, C]$ (where $j \in [d]$) of the loss function is continuous and concave in the second argument.

2. The Learner selects some $x^t \in \mathcal{X}^t$;

3. The Adversary observes the Learner's selection $x^t$, and chooses some action $y^t \in \mathcal{Y}^t$ in response;

4. The Learner's action $x^t \in \Delta\mathcal{A}^t$ is interpreted as a mixture over the pure actions in $\mathcal{A}^t$, and an *outcome* $a^t \in \mathcal{A}^t$ is sampled from it; that is, $a^t \sim x^t$.

5. The Learner suffers (and observes) $\ell^t(a^t, y^t)$, the loss vector with respect to the outcome $a^t$.

Thus, the probabilistic setting is simply a specialization of our framework to the case of the Learner's action set being a simplex at each round.

Unlike in the above deterministic setting, where the transcript through any round $t$ was defined as $\{(x^t, y^t)\}_{s=1}^t$, in the present case we define the transcript through round $t$ as

$$\pi^t := \{(a^1, y^1), \ldots, (a^t, y^t)\},$$

that is, the transcript now records the Learner's *realized outcomes* rather than her chosen mixtures at all rounds. Furthermore, we will denote by $\Pi^t$ the set of transcripts through round $t$, for $t \in [T]$.

Now, let us fix any Adversary $\mathrm{Adv}$ (that is, all of the Adversary's decisions through round $T$). With respect to this fixed Adversary, any *algorithm* for the Learner (defined as the collection of the Learner's decision mappings $\{\pi^{t-1} \to \Delta\mathcal{A}^t\}_{t \in [T]}$ for all rounds) induces an ex-ante distribution $\mathcal{P}_{\mathrm{Adv}}$ over the set of transcripts $\Pi^T$.

Now, we give two types of probabilistic guarantees on the performance of Algorithm B.1, namely, an in-expectation bound and a high-probability bound. Both bounds hold for any choice of Adversary $\mathrm{Adv}$, and are *ex-ante with respect to the algorithm-induced distribution $\mathcal{P}_{\mathrm{Adv}}$ over the final transcripts*.

**Theorem B.1** (In-Expectation Bound). *Given $T \geq \ln d$, Algorithm B.1 with learning rate $\eta = \sqrt{\frac{\ln d}{4TC^2}}$ guarantees that ex-ante, with respect to the randomness in the Learner's realized outcomes, the expected AMF regret is bounded as:*

$$\mathbb{E}\left[R^T\right] \leq 4C\sqrt{T \ln d}.$$

As mentioned in Section B.2.1, the proof of Theorem B.1 is much the same as the proofs of Theorem 2.1 and the helper Lemmas 2.1, 2.2, 2.3, with the exception of using Jensen's inequality to switch the order of taking expectations when necessary. We omit further details.

**Theorem B.2** (High-Probability Bound). *Fix any $\delta \in (0, 1)$. Given $T \geq \ln d$, Algorithm B.1 with learning rate $\eta = \sqrt{\frac{\ln d}{4TC^2}}$ guarantees that the AMF regret will satisfy, with ex-ante probability $1 - \delta$ over the randomness in the Learner's realized outcomes,*

$$R^T \leq 8C\sqrt{T \ln\left(\frac{d}{\delta}\right)}.$$

*Proof.* Throughout this proof, we put tildes over random variables to distinguish them from their realized values. For instance, $\tilde{\pi}^t$ is the random transcript through round $t$, while $\pi^t$ is a realization of $\tilde{\pi}^t$. Also, we explicitly specify the dependence of the surrogate loss $L^t$ on the (random or realized) transcript.

Consider the following random process $\{\tilde{Z}^t\}$, defined recursively for $t = 0, 1, \ldots, T$ and adapted to the sequence of random variables $\tilde{\pi}^1, \ldots, \tilde{\pi}^T$. We let $\tilde{Z}^0 := 0$ deterministically, and for $t \in [T]$ we let

$$\tilde{Z}^t := \tilde{Z}^{t-1} + \ln L^t\left(\tilde{\pi}^t\right) - \mathbb{E}_{\tilde{\pi}^t}\left[\ln L^t\left(\tilde{\pi}^t\right)|\tilde{\pi}^{t-1}\right].$$

It is easy to see that for all $t \in [T]$, we have $\mathbb{E}_{\tilde{\pi}^t}\left[\tilde{Z}^t|\tilde{\pi}^{t-1}\right] = \tilde{Z}^{t-1}$, and thus $\{\tilde{Z}^t\}$ is a martingale.

We next show that this martingale has bounded increments. In brief, this follows from $\{\tilde{Z}^t\}$ being defined in terms of the *logarithm* of the surrogate loss.

**Lemma B.1.** *The martingale $\{\tilde{Z}^t\}$ has bounded increments: $|\tilde{Z}^t - \tilde{Z}^{t-1}| \leq 4\eta C$ for all $t \in [T]$.*

*Proof.* It suffices to establish the bounded increments property for an arbitrary realization of the process. Towards this, fix the full transcript $\pi^T$ of the interaction, and consider any round $t \in [T]$.

Recall from the definition of the surrogate loss that

$$L^t(\pi^t) = \sum_{j \in [d]} \exp\left(\eta R_j^{t-1}\left(\pi^{t-1}\right)\right) \cdot \exp\left(\eta\left(\ell_j^t(a^t, y^t) - w_A^t\right)\right).$$

Thus, noting that $\left|\ell_j^t(a^t, y^t) - w_A^t\right| \leq 2C$ for all $j \in [d]$, we have

$$\frac{L^t(\pi^t)}{L^{t-1}(\pi^{t-1})} = \frac{L^t(\pi^t)}{\sum_{j \in [d]} \exp(\eta R_j^{t-1}(\pi^{t-1}))} \in \left[\exp\left(-\eta \cdot 2C\right), \exp\left(\eta \cdot 2C\right)\right].$$

Taking the logarithm yields

$$\left|\ln L^t\left(\pi^t\right) - \ln L^{t-1}(\pi^{t-1})\right| \leq 2\eta C.$$

In fact, this argument shows that $\left|\ln L^t(\pi_r^t) - \ln L^{t-1}(\pi^{t-1})\right| \leq 2\eta C$ for *any* transcript $\pi_r^t$ that equals $\pi^{t-1}$ on the first $t - 1$ rounds. Hence, taking the expectation over $\tilde{\pi}^t$ conditioned on $\pi^{t-1}$, we obtain:

$$\left|\mathbb{E}\left[\ln L^t\left(\tilde{\pi}^t\right)|\pi^{t-1}\right] - \ln L^{t-1}(\pi^{t-1})\right| \leq 2\eta C.$$

To conclude the proof, it now suffices to observe that:

$$
\begin{aligned}
|Z^t - Z^{t-1}| &= \left|\ln L^t\left(\pi^t\right) - \mathbb{E}[\ln L^t\left(\tilde{\pi}^t\right)|\pi^{t-1}]\right| \\
&\leq \left|\ln L^t(\pi^t) - \ln L^{t-1}\left(\pi^{t-1}\right)\right| + \left|\ln L^{t-1}\left(\pi^{t-1}\right) - \mathbb{E}\left[\ln L^t\left(\tilde{\pi}^t\right)|\pi^{t-1}\right]\right| \\
&\leq 2\eta C + 2\eta C = 4\eta C.
\end{aligned}
$$

$\square$

Having established that $\{\tilde{Z}^t\}$ is a martingale with bounded increments, we can now apply the following concentration bound (see e.g. Dubhashi and Panconesi [2009]).

**Fact 2** (Azuma's Inequality). *Fix $\epsilon > 0$. For any martingale $\{\tilde{Z}^t\}_{t=0}^T$ with $|\tilde{Z}^t - \tilde{Z}^{t-1}| \leq \xi$ for $t \in [T]$,*

$$\Pr\left[\tilde{Z}^T - \tilde{Z}^0 \geq \epsilon\right] \leq \exp\left(-\frac{\epsilon^2}{2\xi^2 T}\right).$$

We instantiate this bound for our martingale with $\tilde{Z}^0 = 0$, $\xi = 4\eta C$, and $\epsilon = \xi\sqrt{2T\ln\frac{1}{\delta}} = 4\eta C\sqrt{2T\ln\frac{1}{\delta}}$, and obtain that for any $\delta \in (0,1)$,

$$\tilde{Z}_T \leq 4\eta C\sqrt{2T\ln\frac{1}{\delta}} \quad \text{with prob. } 1-\delta. \tag{1}$$

At this point, let us express $\tilde{Z}^T$ as follows:

$$\tilde{Z}^T = \sum_{t=1}^T \left(\ln L^t(\tilde{\pi}^t) - \mathop{\mathbb{E}}_{\tilde{\pi}^t}\left[\ln L^t(\tilde{\pi}^t)|\tilde{\pi}^{t-1}\right]\right) = \ln L^T(\tilde{\pi}^T) - \ln L^0 - \sum_{t=1}^T\left(\mathop{\mathbb{E}}_{\tilde{\pi}^t}\left[\ln L^t(\tilde{\pi}^t)|\tilde{\pi}^{t-1}\right] - \ln L^{t-1}(\tilde{\pi}^{t-1})\right).$$

Now, with an eye toward bounding the latter sum, observe that for $t \in [T]$,

$$\begin{aligned}
\mathop{\mathbb{E}}_{\tilde{\pi}^t}\left[\ln L^t(\tilde{\pi}^t)|\tilde{\pi}^{t-1}\right] - \ln L^{t-1}(\tilde{\pi}^{t-1}) &\leq \ln\mathop{\mathbb{E}}_{\tilde{\pi}^t}\left[L^t(\tilde{\pi}^t)|\tilde{\pi}^{t-1}\right] - \ln L^{t-1}(\tilde{\pi}^{t-1}) \\
&\leq \ln\left(\left(4\eta^2 C^2 + 1\right)L^{t-1}(\tilde{\pi}^{t-1})\right) - \ln L^{t-1}(\tilde{\pi}^{t-1}) \\
&= \ln(4\eta^2 C^2 + 1) \\
&\leq 4\eta^2 C^2.
\end{aligned}$$

Here, the first step is via Jensen's inequality and the last step is via $\ln(1+x) \leq x$ for $x > -1$. The second step holds since we can show (via reasoning similar to Lemma 2.3) that for any $T \geq \ln d$, at each round $t \in [T]$ Algorithm B.1 with learning rate $\eta = \sqrt{\frac{\ln d}{4TC^2}}$ achieves:

$$\mathop{\mathbb{E}}_{\tilde{\pi}^t}\left[L^t(\tilde{\pi}^t)|\tilde{\pi}^{t-1}\right] \leq (4\eta^2 C^2 + 1)L^{t-1}(\tilde{\pi}^{t-1}).$$

Combining the above observations with Bound 1 and recalling $L^0 = d$ yields, with probability $\geq 1-\delta$,

$$\begin{aligned}
\tilde{Z}_T \leq 4\eta C\sqrt{2T\ln\frac{1}{\delta}} &\iff \ln L^T(\tilde{\pi}^T) - \ln d - \sum_{t=1}^T\left(\mathop{\mathbb{E}}_{\tilde{\pi}^t}[\ln L^t(\tilde{\pi}^t)|\tilde{\pi}^{t-1}] - \ln L^{t-1}(\tilde{\pi}^{t-1})\right) \leq 4\eta C\sqrt{2T\ln\frac{1}{\delta}} \\
&\iff \ln L^T(\tilde{\pi}^T) \leq \ln d + \sum_{t=1}^T\left(\mathop{\mathbb{E}}_{\tilde{\pi}^t}[\ln L^t(\tilde{\pi}^t)|\tilde{\pi}^{t-1}] - \ln L^{t-1}(\tilde{\pi}^{t-1})\right) + 4\eta C\sqrt{2T\ln\frac{1}{\delta}} \\
&\implies \ln L^T(\tilde{\pi}^T) \leq \ln d + 4\eta^2 C^2 T + 4\eta C\sqrt{2T\ln\frac{1}{\delta}}.
\end{aligned}$$

Using the last inequality, with $\eta = \sqrt{\frac{\ln d}{4TC^2}}$, and the fact that $R^T(\tilde{\pi}^T) \leq \frac{L^T(\tilde{\pi}^T)}{\eta}$ (which is easy to deduce via Lemma 2.1), we thus obtain the desired high-probability AMF regret bound. Specifically, with probability $1-\delta$ we have:

$$\begin{aligned}
R^T(\tilde{\pi}^T) \leq \frac{L^T(\tilde{\pi}^T)}{\eta} &\leq \frac{\ln d}{\eta} + 4\eta C^2 T + 4C\sqrt{2T\ln\frac{1}{\delta}} = 2\sqrt{4C^2 T\ln d} + 4C\sqrt{2T\ln\frac{1}{\delta}} \\
&= 4C\sqrt{T}\left(\sqrt{\ln d} + \sqrt{2\ln\frac{1}{\delta}}\right) \leq 4C\sqrt{T}\cdot\sqrt{2}\cdot\sqrt{\ln d + 2\ln\frac{1}{\delta}} \leq 8C\sqrt{T\ln\frac{d}{\delta}}.
\end{aligned}$$

In the last line, we used that $\sqrt{x} + \sqrt{y} \leq \sqrt{2}\sqrt{x+y}$ for $x, y \geq 0$. $\qquad\square$

# C   No-X-Regret: Definitions, Examples, Algorithms, and Proofs

As a warmup, we begin this subsection by carefully demonstrating how to use our framework to derive bounds and algorithms for the very fundamental *external regret* setting. Then, we derive the same types of existential guarantees in the much more general *subsequence regret* setting. We then specialize these subsequence regret bounds into tight bounds for various existing regret notions (such as internal, adaptive, sleeping experts, and multigroup regret). We conclude this subsection by deriving a general no-subsequence-regret algorithm which in turn specializes to an efficient algorithm in all of our applications.

## C.1   Simple Learning From Expert Advice: External Regret

In the classical experts learning setting Littlestone and Warmuth [1994], the Learner has a set of pure actions ("experts") $\mathcal{A}$. At the outset of each round $t \in [T]$, the Learner chooses a distribution over experts $x^t \in \Delta\mathcal{A}$. The Adversary then comes up with a vector of losses $r^t = (r^t_a)_{a \in \mathcal{A}} \in [0, 1]^{\mathcal{A}}$ corresponding to each expert. Next, the Learner samples $a^t \sim x^t$, and experiences loss corresponding to the expert she chose: $r^t_{a^t}$. The Learner also gets to observe the entire vector of losses $r^t$ for that round. The goal of the Learner is to achieve sublinear *external regret* — that is, to ensure that the difference between her cumulative loss and the loss of the best fixed expert in hindsight grows sublinearly with $T$:

$$R^T_{\text{ext}}(\pi^T) := \sum_{t \in [T]} r^t_{a^t} - \min_{j \in \mathcal{A}} \sum_{t \in [T]} r^t_j = o(T).$$

**Theorem C.1.** *Fix a finite pure action set $\mathcal{A}$ for the Learner and a time horizon $T \geq \ln |\mathcal{A}|$. Then, Algorithm B.1 can be instantiated to guarantee that the Learner's expected external regret is bounded as*

$$\mathbb{E}_{\pi^T} \left[ R^T_{\text{ext}} \left( \pi^T \right) \right] \leq 4\sqrt{T \ln |\mathcal{A}|},$$

*and furthermore that for any $\delta \in (0, 1)$, with ex-ante probability $1 - \delta$ over the Learner's randomness,*

$$R^T_{\text{ext}} \left( \pi^T \right) \leq 8 \sqrt{T \ln \frac{|\mathcal{A}|}{\delta}}.$$

*Proof.* We instantiate our probabilistic framework (see Section B.2.1).

*Defining the strategy spaces.*    We define the Learner's pure action set at each round to be the set $\mathcal{A}$, and the Adversary's strategy space to be the convex and compact set $[0, 1]^{|\mathcal{A}|}$, from which the Adversary chooses each round's collection $(r^t_a)_{a \in \mathcal{A}}$ of all actions' losses.

*Defining the loss functions.*    For $d = |\mathcal{A}|$, we define a $d$-dimensional vector valued loss function $\ell^t = (\ell^t_j)_{j \in \mathcal{A}}$, where for every action $j \in \mathcal{A}$, the corresponding coordinate $\ell^t_j : \mathcal{A} \times [0, 1]^{|\mathcal{A}|} \to [-1, 1]$ is given by

$$\ell^t_j(a, r^t) = r^t_a - r^t_j, \quad \text{for } a \in \mathcal{A}, r^t \in [0, 1]^{|\mathcal{A}|}.$$

It is easy to see that $\ell^t_j(a, \cdot)$ is continuous and concave — in fact, linear — in the second argument for all $j, a \in \mathcal{A}$ and $t \in [T]$. Furthermore, its range is $[-C, C]$, for $C = 1$. This verifies the technical conditions imposed by our framework on the loss functions.

*Applying AMF regret bounds.*    We may now invoke Theorem B.1, which implies the following in-expectation AMF regret bound after round $T$ for the instantiation of Algorithm B.1 with the just defined vector losses $(\ell^t)_{t \in [T]}$:

$$\mathbb{E} \left[ \max_{j \in \mathcal{A}} \sum_{t \in [T]} \ell^t_j(a^t, r^t) - \sum_{t \in [T]} w^t_A \right] \leq 4C\sqrt{T \ln d} = 4\sqrt{T \ln |\mathcal{A}|},$$

where recall that $w_A^t$ is the Adversary-Moves-First (AMF) value at round $t$. Connecting the instantiated AMF regret to the Learner's external regret, we get:

$$\mathbb{E}\left[R_{\text{ext}}^T\right] = \mathbb{E}\left[\max_{j\in\mathcal{A}}\sum_{t\in[T]} r_{a^t}^t - r_j^t\right] = \mathbb{E}\left[\max_{j\in\mathcal{A}}\sum_{t\in[T]} \ell_j^t(a^t, r^t)\right] \leq 4\sqrt{T\ln|\mathcal{A}|} + \sum_{t\in[T]} w_A^t.$$

*Bounding the Adversary-Moves-First value.* To obtain the claimed in-expectation external regret bound, it suffices to show that the AMF value at each round $t\in[T]$ satisfies $w_A^t \leq 0$. Intuitively, this holds because if at some round the Learner knew the Adversary's choice of losses $(r_a^t)_{a\in\mathcal{A}}$ in advance, then she could guarantee herself no added loss in that round by picking the action $a\in\mathcal{A}$ with the smallest loss $r_a^t$.

Formally, for any vector of actions' losses $r^t$, define $a_{r^t}^* := \operatorname{argmin}_{a\in\mathcal{A}} r_a^t$, and notice that

$$\min_{a\in\mathcal{A}}\max_{j\in\mathcal{A}} \ell_j^t(a, r^t) \leq \max_{j\in\mathcal{A}} \ell_j^t\left(a_{r^t}^*, r^t\right) = \max_{j\in\mathcal{A}}\left(r_{a_{r^t}^*}^t - r_j^t\right) = \min_{a\in\mathcal{A}} r_a^t - \min_{j\in\mathcal{A}} r_j^t = 0.$$

The third step follows by definition of $a_{r^t}^*$. Hence, the AMF value is indeed nonpositive at each round:

$$w_A^t = \sup_{r^t\in[0,1]^{|\mathcal{A}|}}\min_{a\in\mathcal{A}}\max_{j\in\mathcal{A}} \ell_j^t(a, r^t) \leq 0.$$

This completes the proof of the in-expectation external regret bound. The high-probability external regret bound follows in the same way from Theorem B.2 of Section B.2.1. $\qquad\square$

A bound of $\sqrt{T\ln|\mathcal{A}|}$ is optimal for external regret in the experts learning setting, and so serves to witness the optimality of Theorem 2.1.

In fact, it is easy to demonstrate that in the external regret setting, the generic probabilistic Algorithm B.1 amounts to the well known Exponential Weights algorithm (Algorithm C.1 below) Littlestone and Warmuth [1994]. To see this, note that Algorithm B.1, when instantiated with the above defined loss functions, has the Learner solve the following problem at each round:

$$x^t \in \operatorname*{argmin}_{x\in\Delta\mathcal{A}}\max_{r^t\in[0,1]^{|\mathcal{A}|}}\sum_{j\in\mathcal{A}} \frac{\exp\left(\eta\sum_{s=1}^{t-1}(r_{a^s}^s - r_j^s)\right)}{\sum_{i\in\mathcal{A}}\exp\left(\eta\sum_{s=1}^{t-1}(r_{a^s}^s - r_i^s)\right)} \mathbb{E}_{a\sim x}\left[r_a^t - r_j^t\right],$$

$$= \operatorname*{argmin}_{x\in\Delta\mathcal{A}}\max_{r^t\in[0,1]^{|\mathcal{A}|}}\sum_{j\in\mathcal{A}} \frac{\exp\left(-\eta\sum_{s=1}^{t-1} r_j^s\right)}{\sum_{i\in\mathcal{A}}\exp\left(-\eta\sum_{s=1}^{t-1} r_i^s\right)} \mathbb{E}_{a\sim x}\left[r_a^t - r_j^t\right],$$

$$= \operatorname*{argmin}_{x\in\Delta\mathcal{A}}\max_{r^t\in[0,1]^{|\mathcal{A}|}}\mathbb{E}_{a\sim x, j\sim\text{EW}_\eta(\pi^{t-1})}\left[r_a^t - r_j^t\right],$$

where we denoted the exponential weights distribution as

$$\text{EW}_\eta(\pi^{t-1}) := \left(\frac{\exp\left(-\eta\sum_{s=1}^{t-1} r_j^s\right)}{\sum_{i\in\mathcal{A}}\exp\left(-\eta\sum_{s=1}^{t-1} r_i^s\right)}\right)_{j\in\mathcal{A}} \in \Delta\mathcal{A}.$$

For any choice of $r^t$ by the Adversary, the quantity inside the expectation, $\ell_j^t(a, r^t) = r_a^t - r_j^t$, is *antisymmetric* in $a$ and $j$: that is, $\ell_j^t(a, r^t) = -\ell_a^t(j, r^t)$. Due to this antisymmetry, no matter which $r^t$ gets selected by the Adversary, by playing $a\sim\text{EW}_\eta(\pi^{t-1})$ the Learner obtains

$$\mathbb{E}_{a,j\sim\text{EW}_\eta(\pi^{t-1})}\left[r_a^t - r_j^t\right] = 0,$$

thus achieving the value of the game. It is also easy to see that $x^t = \text{EW}_\eta(\pi^{t-1})$ is the unique choice of $x^t$ that guarantees nonnegative value, hence Algorithm B.1, when specialized to the external regret setting, is *equivalent* to the Exponential Weights Algorithm C.1.

---

**Algorithm C.1:** The Exponential Weights Algorithm with Learning Rate $\eta$

**for** $t = 1, \ldots, T$ **do**

    Sample $a^t$ such that $a^t = j$ with probability proportional to $\exp\left(-\eta\sum_{s=1}^{t-1} r_j^s\right)$, for $j\in\mathcal{A}$.

---

## C.2 Generalization to Subsequence Regret

Here, we present a generalization of the experts learning framework from which we will be able to derive our other applications to no-regret learning problems. There is again a Learner and an Adversary playing over the course of rounds $t \in [T]$. Initially, the Learner is endowed with a finite set of pure actions $\mathcal{A}$. At each round $t$, the Adversary restricts the Learner's set of available actions for that round to some subset $\mathcal{A}^t \subseteq \mathcal{A}$. The Learner plays a mixture $x^t \in \Delta \mathcal{A}^t$ over the available actions. The Adversary responds by selecting a vector of losses $(r_a^t)_{a \in \mathcal{A}} \in [0, 1]^{|\mathcal{A}|}$ associated with the Learner's pure actions. Next, the Learner samples a pure action $a^t \sim x^t$.

Unlike in the standard setting, the Learner's regret will now be measured not just on the entire sequence of rounds $1, 2, \ldots, T$, but more generally on an arbitrary collection $\mathcal{F}$ of *weighted subsequences* $f : [T] \times \mathcal{A} \to [0, 1]$. The understanding is that for any $f \in \mathcal{F}, t \in [T], a \in \mathcal{A}^t$, the quantity $f(t, a)$ is the "weight" with which round $t$ will be included in the subsequence if the Learner's sampled action is $a$ at that round. The Learner does *not* need to know the subsequences ahead of time; instead the Adversary may announce the values $\{f(t, a)\}_{a \in \mathcal{A}^t, f \in \mathcal{F}}$ to the Learner before the corresponding round $t \in [T]$.

**Definition C.1** (Subsequence Regret). *Given a family of functions $\mathcal{F}$, where each $f \in \mathcal{F}$ is a mapping $f : [T] \times \mathcal{A} \to [0, 1]$, chosen adaptively by the Adversary, and a set of finitely many pure actions $\mathcal{A}$ for the Learner, consider a collection of* action-subsequence pairs $\mathcal{H} \subseteq \mathcal{A} \times \mathcal{F}$.

*The Learner's* subsequence regret *after round $T$ with respect to the collection $\mathcal{H}$ is defined by*

$$R_{\mathcal{H}}^T(\pi^T) := \max_{(j,f) \in \mathcal{H}} \sum_{t \in [T]} f(t, a^t) \left( r_{a^t}^t - r_j^t \right),$$

*where $\pi^T = \{(a^t, r^t)\}_{t \in [T]}$ is the transcript of the interaction.*

For intuition, suppose $\mathcal{F} = \{\mathbf{1}\}$, where $\mathbf{1} : [T] \times \mathcal{A} \to [0, 1]$ satisfies $\mathbf{1}(t, a) = 1$ for all $t, a$. That is, the only relevant subsequence is the entire sequence of rounds $1, 2, \ldots, T$. If we then set $\mathcal{H} = \mathcal{A} \times \mathcal{F}$, subsequence regret specializes to the classical notion of (external) regret which was discussed above.

Moreover, we shall require the following condition on $\mathcal{H}$ and the action sets $\{\mathcal{A}^t\}_{t \in [T]}$, which simply asks that at each round, the Learner be responsible for regret only to currently available actions.

**Definition C.2** (No regret to unavailable actions). *A collection of action-subsequence pairs $\mathcal{H}$, paired with action sets $\{\mathcal{A}^t\}_{t \in [T]}$, satisfy the no-regret-to-unavailable-actions property if at each round $t \in [T]$, for every $f \in \mathcal{F}$ such that $(j, f) \in \mathcal{H}$ for some $j \notin \mathcal{A}^t$, it holds that $f(t, a) = 0$ for all $a \in \mathcal{A}^t$.*

It is worth noting that this condition is trivially satisfied whenever the Learner's action set is invariant across rounds ($\mathcal{A}^t = \mathcal{A}$ for all $t$).

**Theorem C.2.** *Consider a sequence of action sets $\{\mathcal{A}^t\}_{t \in [T]}$ for the Learner, a collection $\mathcal{H}$ of action-subsequence pairs, and a time horizon $T \geq \ln |\mathcal{H}|$. If $\mathcal{H}$ and $\{\mathcal{A}^t\}_{t \in [T]}$ satisfy no-regret-to-unavailable-actions, then an appropriate instantiation of Algorithm B.1 guarantees that the Learner's expected subsequence regret is bounded as*

$$\mathbb{E}_{\pi^T} \left[ R_{\mathcal{H}}^T \left( \pi^T \right) \right] \leq 4 \sqrt{T \ln |\mathcal{H}|},$$

*and furthermore, for any $\delta \in (0, 1)$, that with ex-ante probability $1 - \delta$ over the Learner's randomness,*

$$R_{\mathcal{H}}^T \left( \pi^T \right) \leq 8 \sqrt{T \ln \frac{|\mathcal{H}|}{\delta}}.$$

*Proof.* We instantiate our probabilistic framework of Section B.2.1.

*Defining the strategy spaces.* At each round $t$, the Learner's pure strategy set will be $\mathcal{A}^t$, and the Adversary's strategy space will be the convex and compact set $[0, 1]^{|\mathcal{A}|}$.

*Defining the loss functions.* For all action-subsequence pairs $(j, f) \in \mathcal{H}$, we define the corresponding loss $\ell^t_{(j,f)} : \mathcal{A}^t \times [0,1]^{|\mathcal{A}|} \to [-1,1]$ as

$$\ell^t_{(j,f)}(a, r^t) = f(t, a)(r^t_a - r^t_j), \quad \text{for } a \in \mathcal{A}^t, r^t \in [0,1]^{|\mathcal{A}|}.$$

It is easy to see that for all $(j, f) \in \mathcal{H}$ and each $a \in \mathcal{A}^t$, the function $\ell^t_{(j,f)}(a, \cdot)$ is continuous and concave — in fact, linear — in the second argument, as well as bounded within $[-C, C]$ for $C = 1$. Therefore, the technical conditions imposed by our framework on the loss functions are met.

*Bounding the Adversary-Moves-First value.* At each round $t$, the AMF value $w^t_A = 0$. Trivially, $w^t_A \geq 0$, as the Adversary can always set $r^t_a = 0$ for all $a$. Conversely, $w^t_A \leq 0$ as an easy consequence of the no-regret-to-unavailable-actions property. To see this, for any vector of actions' losses $r^t$, define

$$a^*_{r^t} := \operatorname*{argmin}_{a \in \mathcal{A}^t} r^t_a,$$

and notice that

$$
\begin{aligned}
w^t_A &= \sup_{r^t \in [0,1]^{|\mathcal{A}|}} \min_{a \in \mathcal{A}^t} \left( \max_{(j,f) \in \mathcal{H}} \ell^t_{(j,f)}(a, r^t) \right), \\
&= \sup_{r^t \in [0,1]^{|\mathcal{A}|}} \min_{a \in \mathcal{A}^t} \max \left( \max_{(j,f) \in \mathcal{H}: j \in \mathcal{A}^t} \ell^t_{(j,f)}(a, r^t), 0 \right), \quad \text{(no regret to unavailable actions)} \\
&\leq \sup_{r^t \in [0,1]^{|\mathcal{A}|}} \max \left( \max_{(j,f) \in \mathcal{H}: j \in \mathcal{A}^t} \ell^t_{(j,f)}(a^*_{r^t}, r^t), 0 \right), \\
&= \sup_{r^t \in [0,1]^{|\mathcal{A}|}} \max \left( \max_{(j,f) \in \mathcal{H}: j \in \mathcal{A}^t} f(t, a^*_{r^t})(r^t_{a^*_{r^t}} - r^t_j), 0 \right), \\
&\leq \sup_{r^t \in [0,1]^{|\mathcal{A}|}} \max \left( \max_{(j,f) \in \mathcal{H}: j \in \mathcal{A}^t} f(t, a^*_{r^t})(r^t_j - r^t_j), 0 \right), \quad \text{(by definition of } a^*_{r^t}) \\
&= \sup_{r^t \in [0,1]^{|\mathcal{A}|}} \max (0, 0), \\
&= 0.
\end{aligned}
$$

We thus conclude that Theorems B.1 and B.2 apply (with $C = 1$ and all $w^t_A = 0$) to the subsequence regret setting, yielding the claimed in-expectation and high-probability regret bounds. $\qquad \square$

We now instantiate subsequence regret with various choices of subsequence families, in order to get bounds and efficient algorithms for several standard notions of regret from the literature. For brevity, for each notion of regret considered below we only exhibit the existential in-expectation guarantee for that type of regret, and omit the corresponding high-probability bounds (which are all easily derivable from Theorem B.2). We also point out that all in-expectation bounds cited below are efficiently achievable by instantiating, with appropriate loss functions, the no-subsequence regret Algorithms C.2 and C.3 derived in the following Section C.3.

In all no-regret settings discussed below, except for Sleeping Experts, the Learner has a pure and finite action set $\mathcal{A}$ at every round $t \in [T]$; furthermore — as usual — the Adversary's role at each round consists in selecting the vector of per-action losses $(r^t_a)_{a \in \mathcal{A}} \in [0,1]^{|\mathcal{A}|}$.

**Internal and Swap Regret** To introduce the notion of *internal regret* [Foster and Vohra, 1998], consider the following collection $\mathcal{M}_{\text{int}} \subset \mathcal{A}^{\mathcal{A}}$ of mappings from the action set $\mathcal{A}$ to itself. $\mathcal{M}_{\text{int}}$ consists of the identity map $\mu_{\text{id}}$ (such that $\mu_{\text{id}}(a) = a$ for all $a \in \mathcal{A}$), together with all $|\mathcal{A}|(|\mathcal{A}| - 1)$ maps $\mu_{i \to j}$ that pair two particular actions: i.e., $\mu_{i \to j}(i) = j$, and $\mu_{i \to j}(a) = a$ for $a \neq i$. The Learner's internal regret is then defined as

$$R^T_{\text{int}} := \max_{\mu \in \mathcal{M}_{\text{int}}} \sum_{t \in [T]} r^t_{a^t} - r^t_{\mu(a^t)}.$$

In other words, the Learner's total loss is being compared to all possible counterfactual worlds, for $i, j \in \mathcal{A}$, in which whenever the Learner played some action $i$, it got replaced with action $j$ (and other actions remain fixed).

We can reduce the problem of obtaining no-internal-regret to the problem of obtaining no subsequence regret for a simple choice of subsequences. Let us define the following set of subsequences: $\mathcal{F} := \{f_i : i \in \mathcal{A}\}$, where each $f_i$ is defined to be the indicator of the subsequence where the Learner played action $i$ — that is, for all $t \in [T]$, we let $f_i(t, a) = 1_{a=i}$. Then, we let $\mathcal{H} := \mathcal{A} \times \mathcal{F}$. By the in-expectation no-subsequence-regret guarantee, we then have

$$\mathbb{E}\left[\max_{(j,f)\in\mathcal{H}} \sum_{t\in[T]} f(t, a^t)\left(r_{a^t}^t - r_j^t\right)\right] \leq 4\sqrt{T \ln |\mathcal{H}|} = 4\sqrt{2T \ln |\mathcal{A}|},$$

since $|\mathcal{H}| = |\mathcal{A}| \cdot |\mathcal{F}| = |\mathcal{A}|^2$.

But observe that the Learner's internal regret precisely coincides with the just defined instance of subsequence regret:

$$R_{\text{int}}^T = \max_{\mu\in\mathcal{M}_{\text{int}}} \sum_{t\in[T]} r_{a^t}^t - r_{\mu(a^t)}^t = \max_{i,j\in\mathcal{A}} \sum_{t\in[T]:a^t=i} r_i^t - r_j^t = \max_{j\in\mathcal{A}} \max_{f_i:i\in\mathcal{A}} \sum_{t\in[T]} f_i(t, a^t)(r_{a^t}^t - r_j^t)$$

$$= \max_{(j,f)\in\mathcal{H}} \sum_{t\in[T]} f(t, a^t)(r_{a^t}^t - r_j^t).$$

Therefore, we have established the following existential in-expectation internal regret bound:

$$\mathbb{E}\left[R_{\text{int}}^T\right] \leq 4\sqrt{2T \ln |\mathcal{A}|},$$

which is optimal.

The notion of *swap regret*, introduced in Blum and Mansour [2007], is strictly more demanding than internal regret in that it considers strategy modification rules $\mu$ that can perform more than one action swap at a time. Consider the set $\mathcal{M}_{\text{swap}}$ of all $|\mathcal{A}|^{|\mathcal{A}|}$ *swapping rules* $\mu : \mathcal{A} \to \mathcal{A}$. The Learner's swap regret is defined to be the maximum of her regret to all swapping rules:

$$R_{\text{swap}}^T := \max_{\mu\in\mathcal{M}_{\text{swap}}} \sum_{t\in[T]} r_{a^t}^t - r_{\mu(a^t)}^t.$$

The interpretation is that the Learner's total loss is being compared to the total loss of any remapping of her action sequence.

An easy reduction shows that the swap regret is upper-bounded by $|\mathcal{A}|$ times the internal regret. For completeness, we provide the details of this reduction in Appendix C.4. The reduction implies an in-expectation bound of $4|\mathcal{A}|\sqrt{2T \ln |\mathcal{A}|}$ on swap regret, which, compared to the optimal bound of $O(\sqrt{T|\mathcal{A}| \ln |\mathcal{A}|})$ (see Blum and Mansour [2007]), has suboptimal dependence on $|\mathcal{A}|$.

**Adaptive Regret**   In this setting, consider all contiguous time intervals within rounds $1, \ldots, T$, namely, all intervals $[t_1, t_2]$, where $t_1, t_2$ are integers such that $1 \leq t_1 \leq t_2 \leq T$. The Learner's regret on each interval $[t_1, t_2]$ is defined as her total loss over the rounds $t \in [t_1, t_2]$, minus the loss of the best action for that interval in hindsight. The Learner's adaptive regret is then defined to be her maximum regret over all contiguous time intervals:

$$R_{\text{adaptive}}^T := \max_{[t_1,t_2]:1\leq t_1\leq t_2\leq T} \max_{j\in\mathcal{A}} \sum_{t=t_1}^{t_2} r_{a^t}^t - r_j^t.$$

We observe that adaptive regret corresponds to subsequence regret with respect to $\mathcal{H} := \mathcal{A} \times \mathcal{F}$, where $\mathcal{F} := \{f_{[t_1,t_2]} : 1 \leq t_1 \leq t_2 \leq T\}$ is the collection of subinterval indicator subsequences — that is, $f_{[t_1,t_2]}(t, a) := 1_{t_1\leq t\leq t_2}$ for all $t \in [T]$ and $a \in \mathcal{A}$. Observe that $|\mathcal{F}| \leq T^2$, and therefore, the expected regret upper bound for subsequence regret specializes to the following expected adaptive regret bound:

$$\mathbb{E}\left[R_{\text{adaptive}}^T\right] \leq 4\sqrt{T \ln(|\mathcal{A}||\mathcal{F}|)} \leq 4\sqrt{T(\ln |\mathcal{A}| + 2\ln T)}.$$

**Sleeping Experts**  Following Blum and Mansour [2007], we define the sleeping experts setting as follows. Suppose that the Learner is initially given a set of pure actions $\mathcal{A}$, and before each round $t$, the Adversary chooses a subset of pure actions $\mathcal{A}^t \subseteq \mathcal{A}$ available to the Learner at that round — these are known as the "awake experts", and the rest of the experts are the "sleeping experts" at that round.

The Learner's regret to each action $j \in \mathcal{A}$ is defined to be the excess total loss of the Learner during rounds where $j$ was "awake", compared to the total loss of $j$ over those rounds. Formally, the Learner's sleeping experts regret after round $T$ is defined to be

$$R_{\text{sleeping}}^T := \max_{j \in \mathcal{A}} \sum_{t \in [T]: j \in \mathcal{A}^t} r_{a^t}^t - r_j^t.$$

This is clearly an instance of subsequence regret — indeed, we may consider the family of subsequences $\mathcal{F} := \{f_j : j \in \mathcal{A}\}$, where $f_j(t, a) := 1_{j \in \mathcal{A}^t}$ for all $j, a, t$, and let $\mathcal{H} := \{(j, f_j)\}_{j \in \mathcal{A}}$. It is easy to verify that the no-regret-to-unavailable-actions property holds, and thus the guarantees of the subsequence regret setting carry over to this sleeping experts setting. In particular, the following existential in-expectation sleeping experts regret bound holds:

$$\mathbb{E}\left[R_{\text{sleeping}}^T\right] \leq 4\sqrt{T \ln |\mathcal{A}|},$$

which is also optimal in this setting.

**Multi-Group Regret**  We imagine that before each round, the Adversary selects and reveals to the Learner some *context* $\theta^t$ from an underlying feature space $\Theta$. The interpretation is that the Learner's decision at round $t$ will pertain to an individual with features $\theta^t$. Additionally, there is a fixed collection $\mathcal{G} \subset 2^\Theta$, where each $g \in \mathcal{G}$ is interpreted as a (demographic) group of individuals within the population $\Theta$. Here $\mathcal{G}$ may be large and may consist of overlapping groups. The Learner's goal is to minimize regret to each action $a \in \mathcal{A}$ not just over the entire population, but also separately for each population group $g \in \mathcal{G}$. Explicitly, the Learner's multi-group regret after round $T$ is defined to be

$$R_{\text{multi}}^T := \max_{g \in \mathcal{G}} \max_{j \in \mathcal{A}} \sum_{t \in [T]: \theta^t \in g} r_{a^t}^t - r_j^t.$$

It is easy to see that multi-group regret corresponds to subsequence regret with $\mathcal{H} := \mathcal{A} \times \mathcal{F}$, where $\mathcal{F} := \{f_g : g \in \mathcal{G}\}$ is the collection of group indicator subsequences — that is, $f_g(t, a) := 1_{\theta^t \in g}$ for all $t, a$. Here we are taking advantage of the fact that the functions $f$ on which subsequences are defined need not be known to the algorithm ahead of time, and can be revealed sequentially by the Adversary, allowing us to model adversarially chosen contexts. Therefore, multi-group regret inherits subsequence regret guarantees, and in particular, we obtain the following existential in-expectation multi-group regret bound:

$$\mathbb{E}\left[R_{\text{multi}}^T\right] \leq 4\sqrt{T \ln(|\mathcal{A}||\mathcal{G}|)}.$$

Observe that this bound scales only as $\sqrt{\ln |\mathcal{G}|}$ with respect to the number of population groups, which we can therefore take to be exponentially large in the parameters of the problem.

### C.3  Deriving No-Subsequence-Regret Algorithms

We now present a way to specialize Algorithm B.1 to the setting of subsequence regret with no-regret-to-unavailable-actions. At each round, instead of solving a convex-concave problem, the specialized algorithm will only need to solve a polynomial-sized linear program.

---

**Algorithm C.2:** Efficient No Subsequence Regret Algorithm for the Learner

**for** $t = 1, \ldots, T$ **do**

 Learn the current set of feasible actions $\mathcal{A}^t$ (potentially selected by an Adversary).

 Learn the values $f(t, a)$ for every $a \in \mathcal{A}^t$ and $f \in \mathcal{F}$ (potentially selected by an Adversary).

 Solve for $x^t = (x_a^t)_{a \in \mathcal{A}^t} \in \Delta \mathcal{A}^t$ defined by the following linear inequalities for all $a \in \mathcal{A}^t$:

$$x_a^t \sum_{(j,f) \in \mathcal{H}} \exp\left(\eta \sum_{s=1}^{t-1} \ell_{(j,f)}^s(a^s, r^s)\right) f(t, a) - \sum_{j \in \mathcal{A}^t} x_j^t \sum_{f:(a,f) \in \mathcal{H}} \exp\left(\eta \sum_{s=1}^{t-1} \ell_{(a,f)}^s(a^s, r^s)\right) f(t, j) \leq 0$$

 Sample $a^t \sim x^t$.

---

**Theorem C.3.** *Algorithm C.2 implements Algorithm B.1 in the subsequence regret setting, and achieves the same guarantees.*

*Proof.* In parallel to the notation of Algorithm B.1, we define the following set of weights at round $t \in [T]$:

$$\chi^t_{(j,f)} := \frac{1}{Z^t} \exp\left(\eta \sum_{s=1}^{t-1} \ell^s_{(j,f)}(a^s, r^s)\right),$$

where

$$Z^t := \sum_{(j,f) \in \mathcal{H}} \exp\left(\eta \sum_{s=1}^{t-1} \ell^s_{(j,f)}(a^s, r^s)\right).$$

When instantiated with our current set of loss functions, Algorithm B.1 solves the following zero-sum game at round $t \in [T]$, where we denote $\ell^t_{(j,f)}(x, r^t) := \mathbb{E}_{a \sim x}[\ell^t_{(j,f)}(a, r^t)]$:

$$x^t \in \underset{x \in \Delta \mathcal{A}^t}{\operatorname{argmin}} \max_{r^t \in [0,1]^{|\mathcal{A}|}} \sum_{(j,f) \in \mathcal{H}} \chi^t_{(j,f)} \cdot \ell^t_{(j,f)}(x, r^t).$$

By definition of the loss functions in the subsequence regret setting, the objective function is linear in the Adversary's choice of $r^t$. Thus, let us rewrite the objective as a linear combination of $(r^t_a)_{a \in \mathcal{A}^t}$:

$$\sum_{(j,f) \in \mathcal{H}} \chi^t_{(j,f)} \cdot \ell^t_{(j,f)}(x, r^t),$$

$$= \sum_{(j,f) \in \mathcal{H}} \chi^t_{(j,f)} \sum_{a \in \mathcal{A}^t} x_a \cdot f(t,a) \cdot (r^t_a - r^t_j),$$

$$= \sum_{(j,f) \in \mathcal{H}} \sum_{a \in \mathcal{A}^t} r^t_a \cdot x_a \cdot f(t,a) \cdot \chi^t_{(j,f)} - \sum_{(j,f) \in \mathcal{H}} \sum_{a \in \mathcal{A}^t} r^t_j \cdot x_a \cdot f(t,a) \cdot \chi^t_{(j,f)},$$

which, by the no-regret-to-unavailable actions property,

$$= \sum_{a \in \mathcal{A}^t} r^t_a \cdot x_a \sum_{(j,f) \in \mathcal{H}} f(t,a) \cdot \chi^t_{(j,f)} - \sum_{j \in \mathcal{A}^t} r^t_j \sum_{a \in \mathcal{A}^t} x_a \sum_{f:(j,f) \in \mathcal{H}} f(t,a) \cdot \chi^t_{(j,f)},$$

and now, swapping $j$ and $a$ in the second summation,

$$= \sum_{a \in \mathcal{A}^t} r^t_a \cdot x_a \sum_{(j,f) \in \mathcal{H}} f(t,a) \cdot \chi^t_{(j,f)} - \sum_{a \in \mathcal{A}^t} r^t_a \sum_{j \in \mathcal{A}^t} x_j \sum_{f:(a,f) \in \mathcal{H}} f(t,j) \cdot \chi^t_{(a,f)},$$

$$= \sum_{a \in \mathcal{A}^t} r^t_a \underbrace{\left( x_a \sum_{(j,f) \in \mathcal{H}} f(t,a) \cdot \chi^t_{(j,f)} - \sum_{j \in \mathcal{A}^t} x_j \sum_{f:(a,f) \in \mathcal{H}} f(t,j) \cdot \chi^t_{(a,f)} \right)}_{:=c_a(x)}.$$

Thus, the zero-sum game played at round $t$ has objective function $\sum_{a \in \mathcal{A}^t} c_a(x^t) \cdot r^t_a$, where the coefficients $c_a(x^t)$ do not depend on the Adversary's action $r^t$. Recall that this game has value at most $w^t_A = 0$. Hence, $\max_{a \in \mathcal{A}^t} c_a(x^t) \le 0$ for any minimax optimal strategy $x^t$ for the Learner — since otherwise, if some $c_{a'}(x^t) > 0$, the Adversary would get value $c_{a'}(x^t) > 0$ by setting $r^t_{a'} = 1$ and $r^t_a = 0$ for $a \ne a'$. Conversely, by playing $x^t$ such that $\max_{a \in \mathcal{A}^t} c_a(x^t) \le 0$, the Learner gets value $\le 0$, as $r^t_a \ge 0$ for all $a$.

Therefore, the Learner's choice of $x^t$ is minimax optimal if and only if for all $a \in \mathcal{A}^t$,

$$c_a(x^t) \le 0 \iff Z^t \cdot c_a(x^t) \le 0 \iff$$

$$x^t_a \sum_{(j,f) \in \mathcal{H}} f(t,a) \exp\left(\eta \sum_{s=1}^{t-1} \ell^s_{(j,f)}(a^s, r^s)\right) - \sum_{j \in \mathcal{A}^t} x^t_j \sum_{f:(a,f) \in \mathcal{H}} f(t,j) \exp\left(\eta \sum_{s=1}^{t-1} \ell^s_{(a,f)}(a^s, r^s)\right) \le 0.$$

This recovers Algorithm C.2, concluding the proof. $\square$

**Simplification for Action Independent Subsequences**  The above Algorithm C.2 requires solving a linear feasibility problem. This mirrors how existing algorithms for the special case of minimizing internal regret operate (Blum and Mansour [2007]); recall that internal regret corresponds to subsequence regret for a certain collection of $|\mathcal{A}|$ subsequences that depend on the Learner's action in the current round $t$.

By contrast, if all of our subsequence indicators $f \in \mathcal{F}$ are *action independent*, that is, satisfy $f(t, a) = f(t, a')$ for all $a, a' \in \mathcal{A}$ and $t \in [T]$, then it turns out that we can avoid solving a system of linear inequalities: our equilibrium has a closed form. In what follows, we abuse notation and simply write $f(t)$ for the value of the subsequence $f$ at round $t$.

Observe that if each $f \in \mathcal{F}$ is action independent, then we can rewrite our equilibrium characterization in Algorithm C.2 as the requirement that the Learner's chosen distribution $x^t \in \Delta \mathcal{A}^t$ must satisfy, for each $a \in \mathcal{A}^t$ (provided that $f(t) \neq 0$ for at least some $f \in \mathcal{F}$), the following inequality:

$$
\begin{aligned}
x_a^t &\leq \frac{\sum_{j \in \mathcal{A}^t} x_j^t \sum_{f:(a,f) \in \mathcal{H}} f(t) \exp\left(\eta \sum_{s=1}^{t-1} \ell_{(a,f)}^s(a^s, r^s)\right)}{\sum_{(j,f) \in \mathcal{H}} f(t) \exp\left(\eta \sum_{s=1}^{t-1} \ell_{(j,f)}^s(a^s, r^s)\right)}, \\
&= \frac{\sum_{f:(a,f) \in \mathcal{H}} f(t) \exp\left(\eta \sum_{s=1}^{t-1} \ell_{(a,f)}^s(a^s, r^s)\right)}{\sum_{(j,f) \in \mathcal{H}} f(t) \exp\left(\eta \sum_{s=1}^{t-1} \ell_{(j,f)}^s(a^s, r^s)\right)}.
\end{aligned}
$$

Here the equality follows because $x^t \in \Delta \mathcal{A}^t$ is a probability distribution.

We now observe that setting each $x_a^t$ to be its upper bound, for $a \in \mathcal{A}^t$, yields a probability distribution over $\mathcal{A}^t$, which is consequently the unique feasible solution to the above system. Hence, for action independent subsequences, we have a closed-form implementation of Algorithm C.2 that does not require solving a linear feasibility problem:

---

**Algorithm C.3:** An Efficient Learner for Action Independent Subsequences

  **for** $t = 1, \ldots, T$ **do**

    Learn the current set of feasible actions $\mathcal{A}^t$ and the values $f(t)$ for every $f \in \mathcal{F}$ (potentially selected by an Adversary).

    Sample $a^t \sim x^t$, where for all $a \in \mathcal{A}^t$,

$$
x_a^t = \frac{\sum_{f:(a,f) \in \mathcal{H}} f(t) \exp\left(\eta \sum_{s=1}^{t-1} \ell_{(a,f)}^s(a^s, r^s)\right)}{\sum_{(j,f) \in \mathcal{H}} f(t) \exp\left(\eta \sum_{s=1}^{t-1} \ell_{(j,f)}^s(a^s, r^s)\right)}.
$$

---

## C.4 Omitted Reductions between Different Notions of Regret

**Reducing swap regret to internal regret**  We can upper bound the swap regret by reusing the instance of subsequence regret that we defined to capture internal regret. Recall that it was defined as follows. We let $\mathcal{F} := \{f_i : i \in \mathcal{A}\}$, where each $f_i$ is the indicator of the subsequence of rounds where the Learner played action $i$ — that is, for all $t \in [T]$, we let $f(t, a) = 1_{a=i}$. Then, we let $\mathcal{H} := \mathcal{A} \times \mathcal{F}$. We then obtained the in-expectation regret guarantee

$$
\mathbb{E}\left[\max_{(j,f) \in \mathcal{H}} \sum_{t \in [T]} f(t, a^t)\left(r_{a^t}^t - r_j^t\right)\right] \leq 4\sqrt{2T \ln |\mathcal{A}|}.
$$

Returning to swap regret, note that for any fixed swapping rule $\mu : \mathcal{A} \to \mathcal{A}$, we have

$$
\begin{aligned}
\sum_{t \in [T]} r_{a^t}^t - r_{\mu(a^t)}^t &= \sum_{i \in \mathcal{A}} \sum_{t \in [T]:a^t=i} r_{a^t}^t - r_{\mu(i)}^t \\
&\leq \sum_{i \in \mathcal{A}} \max_{j \in \mathcal{A}} \sum_{t \in [T]:a^t=i} r_{a^t}^t - r_j^t \\
&\leq |\mathcal{A}| \max_{i \in \mathcal{A}} \max_{j \in \mathcal{A}} \sum_{t \in [T]:a^t=i} r_{a^t}^t - r_j^t \\
&= |\mathcal{A}| \max_{(j,f) \in \mathcal{H}} \sum_{t \in [T]} f(t, a^t) \left( r_{a^t}^t - r_j^t \right),
\end{aligned}
$$

where in the last line we simply reparametrized the maximum over $i \in \mathcal{A}$ as the maximum over all $f \in \mathcal{F}$. Since the above holds for any $\mu \in \mathcal{M}_{\text{swap}}$, we have

$$
R_{\text{swap}}^t = \max_{\mu \in \mathcal{M}_{\text{swap}}} \sum_{t \in [T]} r_{a^t}^t - r_{\mu(a^t)}^t \leq |\mathcal{A}| \max_{(j,f) \in \mathcal{H}} \sum_{t \in [T]} f(t, a^t) \left( r_{a^t}^t - r_j^t \right),
$$

and therefore, we conclude that there exists an efficient algorithm that achieves expected swap regret

$$
\mathbb{E} \left[ R_{\text{swap}}^T \right] \leq 4 |\mathcal{A}| \sqrt{2T \ln |\mathcal{A}|}.
$$

**Wide-range regret and its connection to subsequence regret**   The wide-range regret setting was first introduced in Lehrer [2003] and then studied, in particular, in Blum and Mansour [2007] and Greenwald and Jafari [2003]. It is quite general, and is in fact equivalent to the subsequence regret setting, up to a reparametrization.

Just as in the subsequence regret setting, imagine there is a finite family of subsequences $\mathcal{F}$, where each $f \in \mathcal{F}$ has the form $f : [T] \times \mathcal{A} \to [0, 1]$. Moreover, suppose there is a finite family $\mathcal{M}$ of *modification rules*. Each modification rule $\mu \in \mathcal{M}$ is defined as a mapping $\mu : [T] \times \mathcal{A} \to \mathcal{A}$, which has the interpretation that if at time $t$, the Learner plays action $a^t$, then the modification rule modifies this action into another action $\mu(t, a^t) \in \mathcal{A}$. Now, consider a collection of modification rule-subsequence pairs $\mathcal{H} \subseteq \mathcal{M} \times \mathcal{F}$. The Learner's wide-range regret with respect to $\mathcal{H}$ is defined as

$$
R_{\text{wide}}^T := \max_{(\mu,f) \in \mathcal{H}} \sum_{t \in [T]} f(t, a^t) \left( r_{a^t}^t - r_{\mu(t,a^t)}^t \right).
$$

It is evident that wide-range regret has subsequence regret (when the Learner's action set $\mathcal{A}^t = \mathcal{A}$ for all $t \in [T]$) as a special case, where each modification rule $\mu \in \mathcal{M}$ always outputs the same action: that is, for all $t, a^t$, we have $\mu(t, a^t) = j$ for some $j \in \mathcal{A}$.

It is also not hard to establish the converse. Indeed, suppose we have an instance of no-wide-range-regret learning with $\mathcal{H} \subseteq \mathcal{M} \times \mathcal{F}$, where $\mathcal{M}$ is a family of modification rules and $\mathcal{F}$ is a family of subsequences. Fix any pair $(\mu, f) \in \mathcal{H}$. Then, let us define, for all $j \in \mathcal{A}$, the subsequence

$$
\phi_j^{(\mu,f)} : [T] \times \mathcal{A} \to [0, 1] \text{ such that } \phi_j^{(\mu,f)}(t, a) := f(t, a) \cdot 1_{\mu(t,a)=j} \text{ for all } t \in [T], a \in \mathcal{A}.
$$

Now, let us instantiate our subsequence regret setting with

$$
\mathcal{H}_{\text{wide}} := \bigcup_{(\mu,f) \in \mathcal{H}} \bigcup_{j \in \mathcal{A}} \left( j, \phi_j^{(\mu,f)} \right).
$$

Observe in particular that $|\mathcal{H}_{\text{wide}}| = |\mathcal{A}| \cdot |\mathcal{H}|$.

Computing the subsequence regret of this family $\mathcal{H}_{\text{wide}}$, we have

$$
R_{\mathcal{H}_{\text{wide}}}^T = \max_{(\mu,f) \in \mathcal{H}} \max_{j \in \mathcal{A}} \sum_{t \in [T]:\mu(t,a^t)=j} f(t, a^t)(r_{a^t}^t - r_j^t).
$$

Now, we have the following upper bound on the wide-range regret:

$$R_{\text{wide}}^T = \max_{(\mu,f)\in\mathcal{H}} \sum_{t\in[T]} f(t,a^t)\left(r_{a^t}^t - r_{\mu(t,a^t)}^t\right)$$

$$= \max_{(\mu,f)\in\mathcal{H}} \sum_{j\in\mathcal{A}} \sum_{t\in[T]:\mu(t,a^t)=j} f(t,a^t)\left(r_{a^t}^t - r_j^t\right)$$

$$\leq \max_{(\mu,f)\in\mathcal{H}} |\mathcal{A}| \max_{j\in\mathcal{A}} \sum_{t\in[T]:\mu(t,a^t)=j} f(t,a^t)\left(r_{a^t}^t - r_j^t\right)$$

$$= |\mathcal{A}| R_{\mathcal{H}_{\text{wide}}}^T.$$

Since our subsequence regret results imply the existence of an algorithm such that $\mathbb{E}\left[R_{\mathcal{H}_{\text{wide}}}^T\right] \leq 4\sqrt{T\ln|H'|} = 4\sqrt{T(\ln|\mathcal{A}| + \ln|\mathcal{H}|)}$, we have the following expected wide-range regret bound:

$$\mathbb{E}\left[R_{\text{wide}}^T\right] \leq 4|\mathcal{A}|\sqrt{T\left(\ln|\mathcal{A}| + \ln|\mathcal{H}|\right)}.$$

## D  Multicalibration: The Algorithm and Full Proofs

**A simple and efficient algorithm for the Learner**  As mentioned in the proof sketch of Theorem 4.1, in the setting of multicalibration, our framework's general Algorithm B.1 has a particularly simple *approximate* version (originally derived in Gupta et al. [2022]) that lets the Learner (almost) match the above bounds on the multicalibration constant $\alpha$. This approximate algorithm is very efficient and has "low" randomization: namely, at each round the Learner plays an explicitly given distribution which randomizes over at most two points in $\mathcal{A}_r$.

---

**Algorithm D.1:** Simple Multicalibrated Learner

**for** $t = 1, \ldots, T$ **do**
  Observe $\theta^t$.
  For each $i \in [n]$, compute:

$$C_{t-1}^i := \sum_{g\in\mathcal{G}:\theta^t\in g} \exp\left(\eta\sum_{s=1}^{t-1} \ell_{i,g,+1}^s(a^s,b^s)\right) - \exp\left(-\eta\sum_{s=1}^{t-1} \ell_{i,g,+1}^s(a^s,b^s)\right).$$

  **if** $C_{t-1}^i > 0$ for all $i \in [n]$ **then**
    Predict $a^t = 1$.
  **else if** $C_{t-1}^i < 0$ for all $i \in [n]$ **then**
    Predict $a^t = 0$.
  **else**
    Find $j \in [n-1]$ such that $C_{t-1}^j \cdot C_{t-1}^{j+1} \leq 0$.
    Define $q^t \in [0,1]$ as follows (using the convention that 0/0 = 1):

$$q^t := \left|C_{t-1}^{j+1}\right| / \left(\left|C_{t-1}^{j+1}\right| + \left|C_{t-1}^j\right|\right).$$

    Sample $a^t = \frac{j}{n} - \frac{1}{rn}$ with probability $q^t$ and $a^t = \frac{j}{n}$ with probability $1 - q^t$.

---

**Theorem D.1.** *Algorithm D.1 achieves the multicalibration guarantees of Theorem 4.1.*

*Proof.* Let us instantiate the generic probabilistic Algorithm B.1 with our current set of loss functions. In parallel with the notation of Algorithm B.1, for any bucket $i$, group $g$ and $\sigma \in \{-1,+1\}$, we define

$$\chi_{i,g,\sigma}^t := \frac{1}{Z^t}\exp\left(\eta\sum_{s=1}^{t-1}\ell_{i,g,\sigma}^s(a^s,b^s)\right),$$

where

$$Z^t := \sum_{i'\in[n],g'\in\mathcal{G},\sigma'=\pm 1}\exp\left(\eta\sum_{s=1}^{t-1}\ell_{i',g',\sigma'}^s(a^s,b^s)\right).$$

In this notation, at each round $t \in [T]$, the Learner has to solve the following zero-sum game:

$$x^t \in \operatorname*{argmin}_{x \in \Delta \mathcal{A}_r} \max_{b \in [0,1]} \mathbb{E}_{a \sim x} \left[ \xi^t(a,b) \right],$$

where we define

$$\xi^t(a,b) := \sum_{i \in [n], g \in \mathcal{G}, \sigma \in \{-1,1\}} \chi^t_{i,g,\sigma} \cdot \ell^t_{i,g,\sigma}(a,b) \quad \text{for } a \in \mathcal{A}_r, b \in [0,1].$$

For any $a$, let $i_a$ denote the unique bucket index $i \in [n]$ such that $a \in B^i_n$. Substituting

$$\ell^t_{i,g,\sigma}(a,b) = \sigma \cdot \mathbb{1}_{\theta^t \in g} \cdot \mathbb{1}_{a \in B^i_n} \cdot (b - a),$$

we see that most terms in the summation disappear, and what remains is precisely

$$\xi^t(a,b) = \sum_{g \in \mathcal{G}: \theta^t \in g} \sum_{\sigma \in \{-1,1\}} \chi^t_{i_a,g,\sigma} \cdot \sigma(b-a) = (b-a) \cdot \frac{C^{i_a}_{t-1}}{Z^t},$$

where $C^{i_a}_{t-1} = Z^t \sum_{g \in \mathcal{G}: \theta^t \in g} \chi^t_{i_a,g,+1} - \chi^t_{i_a,g,-1}$ is as defined in the pseudocode for Algorithm D.1.

Crucially, for any distribution $x$ chosen by the Learner, her attained utility after the Adversary best-responds has a simple closed form. Namely, given any $x$ played by the Learner, we have

$$\max_{b \in [0,1]} \mathbb{E}_{a \sim x} \left[ \xi^t(a,b) \right] = \frac{1}{Z^t} \left( \max_{b \in [0,1]} \left( b \cdot \mathbb{E}_{a \sim x} \left[ C^{i_a}_{t-1} \right] \right) - \mathbb{E}_{a \sim x} \left[ a \cdot C^{i_a}_{t-1} \right] \right),$$

$$= \frac{1}{Z^t} \left( \max \left( \mathbb{E}_{a \sim x} \left[ C^{i_a}_{t-1} \right], 0 \right) - \mathbb{E}_{a \sim x} \left[ a \cdot C^{i_a}_{t-1} \right] \right).$$

With this in mind, the Learner can easily achieve value 0 in the following two cases. When $C^i_{t-1} > 0$ for all $i \in [n]$, playing $a = 1$ deterministically gives: $\max \left( \mathbb{E}_{a \sim x} \left[ C^{i_a}_{t-1} \right], 0 \right) - \mathbb{E}_{a \sim x} \left[ a \cdot C^{i_a}_{t-1} \right] = \mathbb{E}_{a \sim x} \left[ C^{i_a}_{t-1} \right] - \mathbb{E}_{a \sim x} \left[ C^{i_a}_{t-1} \right] = 0$. When $C^i_{t-1} < 0$ for all $i \in [n]$, she can play $a = 0$ deterministically, ensuring that $\max \left( \mathbb{E}_{a \sim x} \left[ C^{i_a}_{t-1} \right], 0 \right) - \mathbb{E}_{a \sim x} \left[ a \cdot C^{i_a}_{t-1} \right] = 0 - 0 = 0$.

In the final case, when there are nonpositive and nonnegative quantities among $\{ C^i_{t-1} \}_{i \in [n]}$, note that there exists an intermediate index $j \in [n-1]$ such that $C^j_{t-1} \cdot C^{j+1}_{t-1} \leq 0$. Then, it is easy to check that $q^t$, as defined in Algorithm D.1, satisfies

$$q^t C^j_{t-1} + (1 - q^t) C^{j+1}_{t-1} = 0.$$

Using this relation, we obtain that when the Learner plays $a^t = \frac{j}{n} - \frac{1}{rn}$ with probability $q^t$ and $a^t = \frac{j}{n}$ with probability $1 - q^t$, she accomplishes value

$$\max_{b \in [0,1]} \mathbb{E}_{a^t} \left[ \xi^t(a^t, b) \right] = \frac{1}{Z^t} \left( \max \left( \mathbb{E} \left[ C^{i_{a^t}}_{t-1} \right], 0 \right) - \mathbb{E} \left[ a^t \cdot C^{i_{a^t}}_{t-1} \right] \right)$$

$$= \frac{1}{Z^t} \left( \max \left( q^t \cdot C^j_{t-1} + (1 - q^t) C^{j+1}_{t-1}, 0 \right) - \left( q^t \left( \tfrac{j}{n} - \tfrac{1}{rn} \right) C^j_{t-1} + (1 - q^t) \tfrac{j}{n} C^{j+1}_s \right) \right)$$

$$= \frac{1}{Z^t} \cdot \frac{1}{rn} C^j_{t-1},$$

and thus, recalling that $C^{t-1}_j = Z^t \sum_{g \in \mathcal{G}: \theta^t \in g} \chi^t_{j,g,+1} - \chi^t_{j,g,-1}$, we obtain

$$\max_{b \in [0,1]} \mathbb{E}_{a^t} \left[ \xi^t(a^t, b) \right] = \frac{1}{rn} \sum_{g \in \mathcal{G}: \theta^t \in g} \chi^t_{j,g,+1} - \chi^t_{j,g,-1} \leq \frac{1}{rn} \sum_{i \in [n], g \in \mathcal{G}, \sigma = \pm 1} \chi^t_{i,g,\sigma} = \frac{1}{rn},$$

where the last line is due to the quantities $\chi_{i,g,\sigma}$ forming a probability distribution.

Therefore, in the language of Section B.2.2, the Learner who uses Algorithm D.1 guarantees herself *achieved AMF value bounds*

$$w^t_{\text{bd}} = \frac{1}{rn} \text{ for } t \in [T].$$

Hence, by Theorem B.3, our (suboptimal) Learner achieves the claimed multicalibration bounds. $\square$

# E   Multicalibeating: Full Statements and Proofs

## E.1   Calibeating a Single Forecaster: Proof of Theorem 4.2

*Proof of Theorem 4.2.* For the exposition of this full proof, we will employ some probabilistic notation that we have not seen in the main Section 4.2. We briefly define it here.

For any subsequence $S \subseteq [T]$ of rounds, $t \sim S$ denotes a uniformly random round in $S$. We denote the empirical distributions of the values of $f, a, (f, a)$ on $S \subseteq [T]$ by $\mathcal{D}^f(S), \mathcal{D}^a(S), \mathcal{D}^{f \times a}(S)$ (or simply $\mathcal{D}^f, \mathcal{D}^a, \mathcal{D}^{f \times a}$ when $S = [T]$). In this notation, we e.g. have $\mathcal{R}^f(\pi^T) = \mathbb{E}_{d \sim \mathcal{D}^f}[\mathrm{Var}_{t \sim S^d}[b^t]]$.

Our quantity of interest, the Brier score $\mathcal{B}^a$ of the Learner's predictions $a$, is inconvenient to handle: indeed, the calibration-refinement decomposition of $\mathcal{B}^a$ is of little utility since the Learner's predictions can take arbitrary real values (in particular, they might all be distinct, in which case the refinement score would be 0, and all of the Brier score would be contained in the calibration error). Instead, we define a convenient surrogate notion of bucketed Brier/calibration/refinement score.

$$\mathcal{K}_n^a(\pi^T) := \frac{1}{T} \sum_{i \in [n]} |S_i| (\bar{a}(S_i) - \bar{b}(S_i))^2.$$

$$\mathcal{R}_n^a(\pi^T) := \frac{1}{T} \sum_{i \in [n]} \sum_{t \in S_i} (b^t - \bar{b}(S_i))^2 = \frac{1}{T} \sum_{i \in [n]} |S_i| \mathrm{Var}_{t \in S_i}[b^t] = \mathbb{E}_{i \sim \mathcal{D}^i} [\mathrm{Var}_{t \sim S_i}[b^t]].$$

$$\mathcal{B}_n^a(\pi^T) := \mathcal{K}_n^a(\pi^T) + \mathcal{R}_n^a(\pi^T).$$

The following lemma shows that as long as $n$ is large enough, the surrogate Brier score is a good estimate of the true Brier score of our predictions (i.e. our squared error).

**Lemma E.1.** $\mathcal{B}^a \leq \mathcal{B}_n^a + \frac{1}{n}$.

*Proof.* We first compute that the original Brier score $\mathcal{B}^a$ equals

$$\mathcal{B}^a := \frac{1}{T} \sum_{t=1}^T (a^t - b^t)^2 = \frac{1}{T} \sum_{i=1}^n \sum_{t \in S_i} (a^t - b^t)^2 = \frac{1}{T} \sum_{i=1}^n |S_i| \sum_{t \in S_i} \frac{1}{|S_i|} (a^t - b^t)^2.$$

The inner sum is the expectation, over the transcript, of $(a^t - b^t)^2$ conditioned on $a^t \in B_n^i$, so we can write:

$$\mathcal{B}^a = \frac{1}{T} \sum_{i=1}^n |S_i| \mathbb{E}_{t \sim S_i} [(a^t - b^t)^2].$$

We can decompose the expected value as:

$$\mathbb{E}_{t \sim S_i} [(a^t - b^t)^2] = (\mathbb{E}_{t \sim S_i} [a^t - b^t])^2 + \mathrm{Var}_{t \sim S_i}[a^t - b^t].$$

By linearity of expectation, the expectation-squared term satisfies:

$$(\mathbb{E}_{t \sim S_i} [a^t - b^t])^2 = (\bar{a}(S_i) - \bar{b}(S_i))^2.$$

Meanwhile, the variance term can be upper bounded using the following fact:

**Fact 3.** *For any random variables $X, Y$:*

$$\mathrm{Var}[X + Y] = \mathrm{Var}[X] + \mathrm{Var}[Y] + 2\mathrm{Cov}(X, Y) \leq \mathrm{Var}[X] + \mathrm{Var}[Y] + 2\sqrt{\mathrm{Var}[X]\mathrm{Var}[Y]}.$$

*where the inequality follows from an application of Cauchy-Schwartz.*

Instantiating $X = a^t$ and $Y = -b^t$, and upper bounding $\sqrt{\mathrm{Var}[X]} \leq \frac{1}{2n}, \sqrt{\mathrm{Var}[Y]} \leq \frac{1}{2}$, we get:

$$\mathrm{Var}_{t \sim S_i}[a^t - b^t] \leq \mathrm{Var}_{t \sim S_i}[a^t] + \mathrm{Var}_{t \sim S_i}[b^t] + 2\sqrt{\mathrm{Var}_{t \sim S_i}[a^t] \mathrm{Var}_{t \sim S_i}[b^t]},$$

$$\leq \frac{1}{(2n)^2} + \mathrm{Var}_{t \sim S_i}[b^t] + \frac{1}{2n},$$

$$\leq \mathrm{Var}_{t \sim S_i}[b^t] + \frac{1}{n}.$$

Putting the above back together gives the desired bound on the difference of $\mathcal{B}^a$ and $\mathcal{B}_n^a$:

$$\mathcal{B}^a = \frac{1}{T} \sum_{i=1}^n |S_i| \underset{t \sim S_i}{\mathbb{E}} [(a^t - b^t)^2],$$

$$\leq \frac{1}{T} \sum_{i=1}^n |S_i| \left( (\bar{a}(S_i) - \bar{b}(S_i))^2 + \underset{t \sim S_i}{\mathrm{Var}} [b^t] + \frac{1}{n} \right),$$

$$= \frac{1}{T} \sum_{i=1}^n |S_i| (\bar{a}(S_i) - \bar{b}(S_i))^2 + \frac{1}{T} \sum_{i=1}^n |S_i| \underset{t \sim S_i}{\mathrm{Var}} [b^t] + \frac{1}{n},$$

$$= \mathcal{K}_n^a + \mathcal{R}_n^a + \frac{1}{n}.$$

$\square$

Having shown that the surrogate Brier score $\mathcal{B}_n^a$ closely approximates the Learner's original score $\mathcal{B}^a$, we can now focus on bounding the calibration and refinement scores associated with $\mathcal{B}_n^a$.

*Calibration:* Our multicalibration condition on $\Theta$ implies that $\frac{|S_i|}{T} |\bar{b}(S_i) - \bar{a}(S_i)| \leq \alpha$ for $i \in [n]$. The calibration score bound then follows directly.

$$\mathcal{K}_n^a = \frac{1}{T} \sum_{i \in [n]} |S_i| (\bar{b}(S_i) - \bar{a}(S_i))^2 \leq \frac{1}{T} \sum_{i \in [n]} |S_i| |\bar{b}(S_i) - \bar{a}(S_i)| \leq \sum_{i \in [n]} \alpha = \alpha n.$$

*Refinement:* We claim that the Learner's surrogate refinement score relates to the refinement score of the forecaster $f$ as follows:

$$\mathcal{R}_n^a \leq \mathcal{R}^f + \alpha n(|D_f| + 1) + \frac{1}{n}.$$

The proof proceeds in two steps, connecting $\mathcal{R}^f$ and $\mathcal{R}^a$ via a quantity we call $\mathcal{R}^{f \times a}$.

**Definition E.1** (Joint Refinement Score)**.**

$$\mathcal{R}^{f \times a} := \underset{d,i \sim \mathcal{D}^{f \times a}}{\mathbb{E}} [\underset{t \sim S_i^d}{\mathrm{Var}} [b^t]] = \frac{1}{T} \sum_{d \in D_f, i \in [n]} |S_i^d| \underset{t \sim S_i^d}{\mathrm{Var}} [b^t].$$

Recall that refinement score, although we defined it for a forecaster, is really a property of a *partition* of the days. It's equally well defined if, instead of partitioning by days on which a forecaster makes a certain forecast, we partition on say, even and odd days, or sunny vs cloudy vs rainy vs snowy days. Or, in the case of Definition E.1, the partition $\{S_i^d\}_{i \in [n], d \in D}$.

First, note that the joint refinement score of $a$ and $f$ is no worse than the refinement score of $f$.

**Observation 1.** $\mathcal{R}^f \geq \mathcal{R}^{f \times a}$.

Intuitively this should make sense, since $\{S_i^d\}$ is a *refinement* of $f$'s level sets by $a$'s level sets. If $a$ is "useful", then this inequality would be strict, as combining with $a$ would explain away more of the variance. Refining by $a$ cannot decrease the amount of variance captured by the partition.

Reversing our perspective, we can think of $\{S_i^d\}$ as a refinement of $a$'s level sets by $f$'s level sets. The key idea is to use multicalibration to show that refining by $f$ is not "useful." Multicalibration ensures us that almost all of $f$'s explanatory power is captured by $a$.

**Observation 2.** $\mathcal{R}_n^a = \mathcal{R}^{f \times a} + \mathbb{E}_{i \sim \mathcal{D}^a}[\mathrm{Var}_{d \sim \mathcal{D}^f(S_i)}[\bar{b}(S_i^d)]]$.

**Observation 3.** *The extra error term is small:* $\mathbb{E}_{i \sim \mathcal{D}^a}[\mathrm{Var}_{d \sim \mathcal{D}^f(S_i)}[\bar{b}(S_i^d)]] \leq \alpha n(|D| + 1) + \frac{1}{n}$.

Combining these three observations will give us our desired refinement score bound:

$$\mathcal{R}_n^a(b) \leq \mathcal{R}^f + \alpha n(|D| + 1) + \frac{1}{n}.$$

We therefore now prove these observations one by one.

*Proof of Observation 1.* We recall the following fact from probability:

**Fact 4** (Law of Total Variance). *For any random variables $W, Z : \Omega \to \mathbb{R}$ in a probability space,*

$$\mathrm{Var}[Z] = \mathbb{E}[\mathrm{Var}[Z|W]] + \mathrm{Var}[\mathbb{E}[Z|W]].$$

*In particular, since variance is always non-negative:*

$$\mathrm{Var}[Z] \geq \mathbb{E}[\mathrm{Var}[Z|W]].$$

For each fixed $d$, we instantiate this fact with $\Omega = S^d$ (equipped with the discrete $\sigma$-algebra and uniform distribution). $Z(t) := b^t$ and $W(t) := i_{a^t}$, the unique $i$ s.t. $a^t \in B_n^i$. This gives us:

$$\mathrm{Var}_{t \sim S^d}[b^t] \geq \mathbb{E}_{i \sim \mathcal{D}^a(S^d)} \mathrm{Var}_{t \sim S^d}[b^t | a^t \in B_n^i]] = \mathbb{E}_{i \sim \mathcal{D}^a(S^d)} \mathrm{Var}_{t \sim S_i^d}[b^t]].$$

Since this is true for all $d$, the inequality continues to hold in expectation over the $d$'s:

$$\mathcal{R}^f = \mathbb{E}_{d \sim \mathcal{D}^f} \mathrm{Var}_{t \sim S^d}[b^t]] \geq \mathbb{E}_{d, i \sim \mathcal{D}^{f \times a}} \mathrm{Var}_{t \sim S_i^d}[b^t]] = \mathcal{R}^{f \times a}.$$

$\square$

*Proof of Observation 2.* Recall the definition of bucketed refinement:

$$\mathcal{R}_n^a = \mathbb{E}_{i \sim \mathcal{D}^a} \mathrm{Var}_{t \sim S_i}[b^t]].$$

To relate this back to $\mathcal{R}^{f \times a}$, we instantiate Fact 5 again, but flipping the roles of $f$ and $a$: we take the underlying spaces to be the sequences $S_i$ defined by calibrated buckets, and let $W$, the variable we condition on, be the level sets of $f$.

For any fixed $i$ representing a level set of $a$, Fact 5 tells us:

$$\mathrm{Var}_{t \sim S_i}[b^t] = \mathbb{E}_{d \sim \mathcal{D}^f(S_i)} \mathrm{Var}_{t \sim S_i}[b^t | f^t = d]] + \mathrm{Var}_{d \sim \mathcal{D}^f(S_i)} \mathbb{E}_{t \sim S_i}[b^t | f^t = d]] = \mathbb{E}_{d \sim \mathcal{D}^f(S_i)} \mathrm{Var}_{t \sim S_i^d}[b^t]] + \mathrm{Var}_{d \sim \mathcal{D}^f(S_i)}[\bar{b}(S_i^d)].$$

Like before, we take the expectation over all $i \in [n]$, giving us the desired result:

$$\mathcal{R}_n^a = \mathbb{E}_{i \sim \mathcal{D}^a} \mathrm{Var}_{t \sim S_i}[b^t]] = \mathbb{E}_{d, i \sim \mathcal{D}^{f \times a}} \mathrm{Var}_{t \sim S_i^d}[b^t]] + \mathbb{E}_{i \sim \mathcal{D}^a} \mathrm{Var}_{d \sim \mathcal{D}^f(S_i)}[\bar{b}(S_i^d)]] = \mathcal{R}^{f \times a} + \mathbb{E}_{i \sim \mathcal{D}^a} \mathrm{Var}_{d \sim \mathcal{D}^f(S_i)}[\bar{b}(S_i^d)]].$$

$\square$

*Proof of Observation 3.* We have to bound the extra error term:

$$\mathbb{E}_{i \sim \mathcal{D}^a} \mathrm{Var}_{d \sim \mathcal{D}^f(S_i)}[\bar{b}(S_i^d)]].$$

In words, this is the expected variance of the true averages on $S_i^d$, conditioned on the buckets $i$. Intuitively, if these true averages vary a lot, then the calibration error on the $S_i^d$s must be large since the prediction on each of the $S_i^d$s is (close to) $i/n$; in particular, they are almost constant across $d$.

Conversely, if multicalibration error is low, then the variance must be low as well. Formally,

$$\mathbb{E}_{i \sim \mathcal{D}^a} [\underset{d \sim \mathcal{D}^f(S_i)}{\mathrm{Var}} [\bar{b}(S_i^d)]] = \sum_{i \in [n]} \frac{|S_i|}{T} (\underset{d}{\mathrm{Var}} [\underset{t \sim S_i^d}{\mathbb{E}} [b^t]]),$$

$$= \sum_{i \in [n]} \frac{|S_i|}{T} (\sum_{d \in D} \frac{|S_i^d|}{|S_i|} (\bar{b}(S_i^d) - \bar{b}(S_i))^2),$$

$$= \sum_{i,d} \frac{|S_i^d|}{T} (\bar{b}(S_i^d) - \bar{b}(S_i))^2,$$

$$\leq \sum_{i,d} \frac{|S_i^d|}{T} |\bar{b}(S_i^d) - \bar{b}(S_i)|,$$

$$\leq \sum_{i,d} \frac{|S_i^d|}{T} (|\bar{b}(S_i^d) - \bar{a}(S_i^d)| + |\bar{a}(S_i^d) - \bar{a}(S_i)| + |\bar{a}(S_i) - \bar{b}(S_i)|),$$

$$\leq \sum_{i,d} \frac{|S_i^d|}{T} (T\alpha/|S_i^d| + \frac{1}{n} + T\alpha/|S_i|),$$

$$\leq \frac{1}{n} + \sum_i \alpha + \sum_{i,d} \alpha,$$

$$= \frac{1}{n} + \alpha n(|D| + 1).$$

The first line is just expanding out the definition. In the third line, we upperbound square with absolute value, since all values are at most 1. In the forth line, we break apart the error term into the difference between our average prediction on $S_i^d$ and the true average (upperbounded by $T\alpha/|S_i^d|$, by calibration guarantees w.r.t $\mathcal{S}(f)$), the difference between our prediction on $S_i^d$ and our average prediction on $S_i$ (which is upper bounded by $1/n$, the size of our bucketing), and the difference between our average prediction on $S_i$ and the true average (upperbounded by $T\alpha/|S_i|$). $\qquad\square$

We have shown that $\mathcal{K}_n^a \leq \alpha n$, and our three observations have given us that $\mathcal{R}_n^a(b) \leq \mathcal{R}^f + \alpha n(|D| + 1) + \frac{1}{n}$. Combining these results and Lemma E.1, we obtain the desired bound: $\mathcal{B}^a \leq \mathcal{R}^f + \alpha n(|D| + 2) + \frac{2}{n}$. This concludes the proof of Theorem 4.2. $\qquad\square$

## E.2 Applying Theorem 4.2: Explicit Rates and Multiple Forecasters

First, we show how to instantiate Theorem 4.2 with our efficiently achievable multicalibration guarantees on $\alpha$ of Theorem 4.1.

**Corollary E.1.** *When run with parameters $r, n \geq 1$ on the collection $\mathcal{G}' := \mathcal{S}(f) \cup \{\Theta\}$, the multicalibration algorithm (Algorithm D.1) $\tau$-calibeats $f$, where*

$$\mathbb{E}[\tau] \leq \frac{2}{n} + n(|D_f| + 2) \left( \frac{1}{rn} + 4\sqrt{\frac{\ln(2(|D_f| + 1)n)}{T}} \right),$$

*and for any $\delta \in (0, 1)$, with probability $1 - \delta$,*

$$\tau \leq \frac{2}{n} + n(|D_f| + 2) \left( \frac{1}{rn} + 8\sqrt{\frac{1}{T} \ln \left( \frac{2(|D_f| + 1)n}{\delta} \right)} \right).$$

*The calibration error overall of the algorithm is bounded, for any $\delta \in (0, 1)$, as:*

$$\mathbb{E}[\mathcal{K}_n^a] \leq \frac{1}{r} + 4n\sqrt{\frac{\ln(2(|D_f| + 1)n)}{T}} \quad and \quad \mathcal{K}_n^a \leq \frac{1}{r} + 8n\sqrt{\frac{1}{T} \ln \left( \frac{2(|D_f| + 1)n}{\delta} \right)} \quad w. \; prob. \; 1 - \delta.$$

*Proof.* Using our online multicalibration guarantees, we get (by Theorem D.1):

$$\mathbb{E}[\alpha] \leq \frac{1}{rn} + 4\sqrt{\frac{\ln(2(|D_f| + 1)n)}{T}},$$

and, for any $\delta \in (0, 1)$, with probability $1 - \delta$:

$$\alpha \leq \frac{1}{rn} + 8\sqrt{\frac{1}{T} \ln\left(\frac{2(|D_f| + 1)n}{\delta}\right)},$$

Plugging this into the result from Theorem 4.2:

$$\mathcal{B}^a - \mathcal{R}^f \leq \alpha n(|D| + 2) + \frac{2}{n},$$

we obtain the desired in-expectation bound on $\tau$:

$$\mathbb{E}[\tau] \leq \frac{2}{n} + n(|D_f| + 2)\,\mathbb{E}[\alpha] \leq \frac{2}{n} + n(|D_f| + 2)\left(\frac{1}{rn} + 4\sqrt{\frac{\ln(2(|D_f| + 1)n)}{T}}\right).$$

We can do so similarly for the high probability bound, so that with probability $1 - \delta$:

$$\tau \leq \frac{2}{n} + n(|D_f| + 2)\left(\frac{1}{rn} + 8\sqrt{\frac{1}{T} \ln\left(\frac{2(|D_f| + 1)n}{\delta}\right)}\right).$$

Finally, the overall calibration error follows directly by plugging in for $\alpha$. $\qquad\square$

The main utility in our approach to calibeating is that it easily extends to multicalibeating. As a warm up, we start by deriving calibration with respect to an ensemble of forecasters. The main result then combines this with calibeating on groups to attain the multicalibeating from Definition 4.5.

**Calibeating an ensemble of forecasters**    Since our result above is based on bounds on multicalibration, we can easily extend it to calibeating an ensemble of forecasters $\mathcal{F}$ by asking for multicalibration with respect to the level sets of all forecasters. More formally, define the groups as:

$$\left(\bigcup_{f \in \mathcal{F}} \mathcal{S}(f)\right) \cup \{\Theta\}.$$

Theorem 4.2 applies separately to each $f$. The only degradation comes in the $\alpha$, since we're asking for multicalibration with respect to more groups. But this effect is small, since the dependence on the number of groups is only $O(\sqrt{\ln|\mathcal{G}|})$.

**Corollary E.2** (Ensemble Calibeating). *On groups $\mathcal{G}' := \left(\bigcup_{f \in \mathcal{F}} \mathcal{S}(f)\right) \cup \{\Theta\}$, the multicalibration algorithm with parameters $r, n \geq 1$, after $T$ rounds attains $(\mathcal{F}, \{\Theta\}, \beta)$-multicalibeating with*

$$\mathbb{E}[\beta(f, \Theta)] \leq \frac{2}{n} + n(|D_f| + 2)\left(\frac{1}{rn} + 4\sqrt{\frac{\ln(2(1 + \sum_{f' \in \mathcal{F}} D_{f'})n)}{T}}\right).$$

*Proof.* We instantiate Theorem D.1 with group collection size $|\mathcal{G}'| = 1 + \sum_{f' \in \mathcal{F}} |D_{f'}|$ to conclude that the multicalibrated algorithm achieves $(\alpha, n)$-multicalibration, with

$$\mathbb{E}[\alpha] \leq \frac{1}{rn} + 4\sqrt{\frac{\ln(2(1 + \sum_{f' \in \mathcal{F}} D_{f'})n)}{T}}.$$

Now, $\forall f \in \mathcal{F} : \mathcal{S}(f) \cup \{\Theta\} \subseteq \mathcal{G}'$ for every $f \in \mathcal{F}$, so we can instantiate Theorem 4.2 for every forecaster $f \in \mathcal{F}$ to give us:

$$\mathcal{B}^a - \mathcal{R}^f \leq \alpha n(|D_f| + 2) + \frac{2}{n} \quad \forall f \in \mathcal{F}.$$

Plugging in the in-expectation bound on $\alpha$, we conclude:

$$\mathbb{E}[\beta(f, \Theta)] \leq \mathbb{E}[\alpha] \cdot n(|D_f| + 2) + \frac{2}{n} \leq \frac{2}{n} + n(|D_f| + 2)\left(\frac{1}{rn} + 4\sqrt{\frac{\ln(2(1 + \sum_{f' \in \mathcal{F}} D_{f'})n)}{T}}\right).$$

$\qquad\square$

### E.3 Multicalibeating + Multicalibration Theorem 4.3: Full Statement and Proof

Recall that for every group $g \in \mathcal{G}$, we let $S(g)$ denote the subsequence of days on which $g$ occurs, where the transcript is left implicit.

**Theorem E.1** (Multicalibeating + Multicalibration: Full version with high-probability bounds)**.** *Let $\mathcal{G} \subseteq 2^{\Theta}$, and $\mathcal{F}$ some set of forecasters $f : \Theta \to D_f$. The multicalibration algorithm on $\mathcal{G}' := \left( \bigcup_{f \in \mathcal{F}} \{ g \cap S : (g, S) \in \mathcal{G} \times \mathcal{S}(f) \} \right) \cup \mathcal{G}$ with parameters $r, n \geq 1$, after $T$ rounds, attains expected $(\mathcal{F}, \mathcal{G}, \beta)$-multicalibeating, where:* [E.1]

$$\mathbb{E}[\beta(f,g)] \leq \frac{2}{n} + \frac{|D_f| + 2}{r \cdot |S(g)|/T} + 4n(|D_f| + 2)\sqrt{\frac{1}{|S(g)|^2/T} \ln\left(2n|\mathcal{G}|(1 + \sum_f |D_f|)\right)} \ \forall \ f \in \mathcal{F}, g \in \mathcal{G},$$

*while maintaining $(\alpha, n)$-multicalibration on the original collection $\mathcal{G}$, with:*

$$\mathbb{E}[\alpha] \leq \frac{1}{rn} + 4\sqrt{\frac{1}{T} \ln\left(2n|\mathcal{G}|(1 + \sum_f |D_f|)\right)}.$$

*We also have the corresponding high probability bounds. For any $\delta \in (0,1)$, with probability $1 - \delta$:*

$$\beta(f,g) \leq \frac{2}{n} + \frac{|D_f| + 2}{r \cdot |S(g)|/T} + 8n(|D_f| + 2)\sqrt{\frac{1}{|S(g)|^2/T} \ln\left(\frac{2n|\mathcal{G}|(1 + \sum_f |D_f|)}{\delta}\right)} \ \forall \ f \in \mathcal{F}, g \in \mathcal{G},$$

*and on the original collection $\mathcal{G}$, the multicalibration constant $\alpha$ satisfies, with probability $1 - \delta$,*

$$\alpha \leq \frac{1}{rn} + 8\sqrt{\frac{1}{T} \ln\left(\frac{2n|\mathcal{G}|(1 + \sum_f |D_f|)}{\delta}\right)}.$$

*Proof.* We begin with a preliminary observation that translates our overall multicalibration assumptions into guarantees over the individual sequences $S(g)$, for $g \in \mathcal{G}$.

**Observation 4.** *Let $a$ be $(\alpha, n)$-multicalibrated on groups $\mathcal{G}'$ over the entire time sequence $[T]$. Then, for any $g$, on the subsequence of days $S(g)$ the predictor $a$ is $\left(\alpha \frac{T}{|S(g)|}, n\right)$-multicalibrated with respect to groups $\left( \bigcup_{f \in \mathcal{F}} \mathcal{S}(f) \right) \cup \{\Theta\}$.*

*Proof.* Let $g \in \mathcal{G}$ be some particular group. Also, fix any $f \in \mathcal{F}$ and $S \in \mathcal{S}(f) \cup \{\Theta\}$. Using multicalibration guarantees (Definition 4.1), we have that for every $i \in [n]$:

$$\left| \sum_{t \in S(g): \, \theta^t \in S \text{ and } a^t \in B_n^i} b^t - a^t \right| = \left| \sum_{t \in [T]: \, \theta^t \in g \cap S \text{ and } a^t \in B_n^i} b^t - a^t \right| \leq \alpha T = \left( \alpha \frac{T}{|S(g)|} \right) |S(g)|.$$

The first equality is by definition of $S(g)$; in particular, $\theta^t \in g \cap S \iff t \in S(g) \wedge \theta^t \in S$. This concludes the proof of our observation. $\square$

With this observation in hand, the proof is again a direct application of Theorem 4.2.

We can instantiate Theorem D.1 with groups $\mathcal{G}'$ to conclude that the multicalibrated algorithm achieves $(\alpha, n)$-multicalibration, with (choosing any $\delta \in (0,1)$):

$$\mathbb{E}[\alpha] \leq \frac{1}{rn} + 4\sqrt{\frac{\ln(2|\mathcal{G}'|n)}{T}}, \quad \text{and} \quad \alpha \leq \frac{1}{rn} + 8\sqrt{\frac{1}{T} \ln\left(\frac{2|\mathcal{G}'|n}{\delta}\right)} \text{ w. prob. } 1 - \delta.$$

where $|\mathcal{G}'| = |\mathcal{G}| + |\mathcal{G}|(\sum_f D_f) = |\mathcal{G}|(1 + \sum_f D_f)$.

---

[E.1] $S(g)$ denotes the subsequence of days on which a group $g$ occurs, suppressing dependence on transcript.

Now, fix any $g \in \mathcal{G}$ and $f \in \mathcal{F}$. By our observation above, we are $\alpha \frac{T}{|S(g)|}$ multicalibrated w.r.t. $S(f) \cup \{\Theta\}$ on the sequence of days on which $g$ occurs. Therefore, we can instantiate Theorem 4.2:

$$\mathcal{B}^a(\pi^T|_{\{t:\theta^t \in g\}}) - \mathcal{R}^f(\pi^T|_{\{t:\theta^t \in g\}}) \le \frac{2}{n} + n(|D_f| + 2)\,\alpha\,\frac{T}{|S(g)|}.$$

Inserting the above bounds on $\alpha$ yields our in-expectation and high-probability bounds on $\beta(\cdot, \cdot)$.

Additionally, the theorem posits that the predictor is also $(\alpha, n)$-multicalibrated on the base collection of subgroups $\mathcal{G}$. Indeed, we have included the family $\mathcal{G}$ into the collection $\mathcal{G}'$, hence the predictor will be $(\alpha, n)$-multicalibrated on $\mathcal{G}$. $\qquad\square$

# F    Blackwell Approachability: The Algorithm and Full Proofs

*Proof of Theorem 5.1.* We instantiate our probabilistic framework of Section B.2.1. The Learner's and Adversary's action sets are inherited from the underlying Polytope Blackwell game.

*Defining the loss functions.*    For all $t = 1, 2, \ldots$, we consider the following losses:

$$\ell^t_{h_{\alpha,\beta}}(x, y) := \langle \alpha, u(x, y) \rangle - \beta, \quad \text{for } h_{\alpha,\beta} \in \mathcal{H}, x \in \mathcal{X}, y \in \mathcal{Y},$$

where here and below the notational convention is that for $x \in \mathcal{X}, y \in \mathcal{Y}$, $u(x, y) := \mathbb{E}_{a \sim x}[u(a, y)]$. The coordinates of the resulting vector loss $\ell^t_{\mathcal{H}}(x, y) := \left(\ell^t_{h_{\alpha,\beta}}(x, y)\right)_{h_{\alpha,\beta} \in \mathcal{H}}$ correspond to the collection $\mathcal{H}$ of the halfspaces that define the polytope. By Holder's inequality, each vector loss function $\ell^t_{\mathcal{H}} \in [-2, 2]^d$ — this follows because we required that for some $p, q$ with $\frac{1}{p} + \frac{1}{q} = 1$, the family $\mathcal{H}$ is $p$-normalized, and the range of $u$ is contained in $B^d_q$. In addition, each $\ell^t_{h_{\alpha,\beta}}$ is continuous and convex-concave, as it is a linear transformation of the continuous and affine-concave function $u$.

*Bounding the Adversary-Moves-First value.*    We observe that for $t \in [T]$, the AMF value $w^t_A \le 0$. Indeed, if the Adversary moves first and selects any $y^t \in \mathcal{Y}$, then by the assumption of response satisfiability, the Learner has some $x^t \in \mathcal{X}$ guaranteeing that $u(x^t, y^t) \in P(\mathcal{H})$. The latter is equivalent to $\ell^t_{h_{\alpha,\beta}}(x^t, y^t) = \langle \alpha, u(x^t, y^t) \rangle - \beta \le 0$ for all $h_{\alpha,\beta} \in \mathcal{H}$, letting us conclude that for any round $t$,

$$w^t_A = \sup_{y^t \in \mathcal{Y}} \min_{x^t \in \mathcal{X}} \left( \max_{h_{\alpha,\beta} \in \mathcal{H}} \ell^t_{h_{\alpha,\beta}}(x^t, y^t) \right) \le 0.$$

*Applying AMF regret bounds.*    Given this instantiation of our framework, Theorem B.1 implies that for any response satisfiable Polytope Blackwell game, the Learner can use Algorithm B.1 (instantiated with the above loss functions) to ensure that after any round $T \ge \ln |\mathcal{H}|$,

$$\mathbb{E}\left[ \max_{h_{\alpha,\beta} \in \mathcal{H}} \sum_{t \in [T]} \left( \langle \alpha, u\left(a^t, y^t\right) \rangle - \beta \right) \right] \le \mathbb{E}\left[ \max_{h_{\alpha,\beta} \in \mathcal{H}} \sum_{t \in [T]} \ell^t_{h_{\alpha,\beta}}(a^t, y^t) - \sum_{t=1}^{T} w^t_A \right] \le 8\sqrt{T \ln |\mathcal{H}|},$$

where the expectation is with respect to the Learner's randomness. Given this guarantee, we obtain, using the definition of $\bar{u}^T$, that

$$\max_{h_{\alpha,\beta} \in \mathcal{H}} \mathbb{E}\left[ \langle \alpha, \bar{u}^T \rangle - \beta \right] \le 8\sqrt{\frac{\ln |\mathcal{H}|}{T}}.$$

Using $T = T(\epsilon) \ge \ln |\mathcal{H}|$, we have that for every $h_{\alpha,\beta} \in \mathcal{H}$,

$$\mathbb{E}\left[ \langle \alpha, \bar{u}^{T(\epsilon)} \rangle - \beta \right] \le 8\sqrt{\frac{\ln |\mathcal{H}|}{T(\epsilon)}} = 8\sqrt{\frac{\ln |\mathcal{H}|}{64 \ln |\mathcal{H}|/\epsilon^2}} = \epsilon.$$

This concludes the proof of our in-expectation guarantee for Polytope Blackwell games.

The high-probability statement follows directly from Theorem B.2, using $C = 2$. $\qquad\square$

**An LP based algorithm when the Adversary has a finite pure strategy space.** Algorithm B.1, which achieves the guarantees of Theorem 5.1, generally involves solving a convex program at each round. It is worth pointing out that only a *linear program* will need to be solved at each round in the commonly studied special case of Blackwell approachability where *both* the Learner and the Adversary randomize between actions in their respective finite action sets $\mathcal{A}$ and $\mathcal{B}$.

Formally, in the setting above, suppose additionally that the Adversary's action space is $\mathcal{Y} = \Delta\mathcal{B}$, where $\mathcal{B}$ is a finite set of pure actions for the Adversary. At each round $t$, *both* the Learner and the Adversary randomize over their respective action sets. First, the Learner selects a mixture $x^t \in \Delta\mathcal{A}$, and then the Adversary selects a mixture $y^t \in \Delta\mathcal{B}$ in response. Next, pure actions $a^t \sim x^t$ and $b^t \sim y^t$ are sampled from the chosen mixtures, and the vector valued utility in that round is set to $u(a^t, b^t)$.

In this fully probabilistic setting, at each round $t$ Algorithm B.1 has the Learner solve a normal-form zero-sum game with pure action sets $\mathcal{A}, \mathcal{B}$, where the utility to the Adversary (the max player) is

$$\xi^t(a, b) := \sum_{h_{\alpha,\beta} \in \mathcal{H}} \exp\left(\eta \sum_{s=1}^{t-1} (\langle \alpha, u(a^s, b^s)\rangle - \beta)\right) \cdot (\langle \alpha, u(a, b)\rangle - \beta) \text{ for } a \in \mathcal{A}, b \in \mathcal{B}. \quad (2)$$

A standard LP-based approach to solving this zero-sum game (see e.g. Raghavan [1994]) is for the Learner to select among distributions $x^t \in \Delta\mathcal{A}$ with the goal of minimizing the maximum payoff to the Adversary over all pure responses $b \in \mathcal{B}$. Writing this down as a linear program, we obtain the following algorithm:

---

**Algorithm F.1:** Linear Programming Based Learner for Polytope Blackwell Approachability

---

**for** $t = 1, \ldots, T$ **do**

    Choose a mixture $x^t = (x_a^t)_{a \in \mathcal{A}} \in \Delta\mathcal{A}$ that solves the following linear program (where $\xi^t(\cdot, \cdot)$ is defined in (2), and $z$ is an unconstrained variable):

$$\text{Minimize } z$$
$$\text{s.t. } \forall b \in \mathcal{B} : \quad z \geq \sum_{a \in \mathcal{A}} x_a^t \, \xi^t(a, b).$$

    Sample $a^t \sim x^t$.

---