# OpenReview forum: "Online Minimax Multiobjective Optimization: Multicalibeating and Other Applications"
_NeurIPS.cc/2022/Conference — NeurIPS 2022 Accept_

### Official Review · Reviewer_x4kU · 2022-07-07

**Rating:** 9
**Confidence:** 2
**Soundness:** 4 excellent
**Presentation:** 4 excellent
**Contribution:** 3 good

**Summary:**

This paper introduces a simple and general online learning framework with adversary plays. And the author shows how to (multi)calibeat and multicalibrate at the same time.

**Questions:**

Should you add some experiments to the main paper?

**Ethics Review Area:**

["I don’t know"]

**Limitations:**

line 228, introduces.


**Strengths And Weaknesses:**

Generally, this work is well-written and solid.
My main concern is that this paper seems to be too heavy for a conference paper. It includes too many ingredients such as softmax, the main conclusion, applications, and extensions.
I guess they should separate it into several papers and only focus on one term each time.

---

> ### Author Response · Authors · 2022-08-01
> **Reply to reviewer x4kU**
>
> Thank you for your review! We agree with you that the manuscript is long. However, in our view, showing that so many different problems can all be easily solved within the same simple framework is one of the most interesting parts of our contribution, and so we think it is important to keep all of the applications within the same paper. In our opinion, splitting this paper up by application would obfuscate this message. We are, however, reorganizing it (as suggested by Reviewer 2) to have simpler applications first so as to make the paper smoother to read.
>
> Especially given its length, we don't think this paper needs experimental evaluation. The main contribution of our paper is not primarily any of the particular algorithms (many of which existed in one form or another already), but rather their common derivation.

---

### Official Review · Reviewer_mPvY · 2022-07-08

**Rating:** 8
**Confidence:** 3
**Soundness:** 4 excellent
**Presentation:** 3 good
**Contribution:** 4 excellent

**Summary:**

The paper makes several theoretical contributions to multi-objective online learning and its applications. First, it proposes a new performance metric called "Adversary-Moves-First" regret, and a novel algorithm to control it via solving a convex-concave problem in each round. This general framework leads to three interesting applications:

1. For the expert problem, it yields an algorithm with sublinear "subsequence regret", which subsumes many well-known performance metrics. Moreover, special versions of this algorithm recover classical solutions designed from first principles, like EW.
2. It leads to an approachability algorithm for polytopes, whose approaching rate depends logarithmically on the amount of constraints. This recovers a strong result from the approachability literature.
3. For the calibration problem, it improves a recent result on "calibeating": the bound depends logarithmically, rather than polynomially, on the number of forecasters. Moreover, this can be combined with the "multicalibration" task to achieve a stronger goal called "multicalibeating". The proof strategy improves an existing level set analysis.

**Questions:**

1. There seems to be an intriguing connection between the proposed multi-objective setting and the standard expert problem (Appendix F.1), and this intuitive similarity extends to the algorithms: Similar to EW, the analysis in Section 2.2 also argues that the log-sum-exp potential does not grow too fast, despite a quite significant deviation in Lemma 2.3. It might be helpful to add some discussions in Section 2.2 and emphasize the novelty of this reasoning compared to EW.
2. From a somewhat aesthetic point of view, although the EW algorithm is recovered in Appendix F.1, the constant $4$ in the bound is not the asymptotically optimal one achieved by EW ($1/\sqrt{2}$). Improving this constant might make the general framework even more appealing.
3. Based on the connection to the expert problem, I think the proposed general algorithm might be interpreted as a regret-computation tradeoff. With a known $T$, we can use dynamic programming to obtain the absolutely optimal strategy for the expert problem. Apparently this is not computationally feasible, therefore EW serves as a proximal solution, which is fast and asymptotically optimal.\
As for the multi-objective setting, if we fix the domains then DP can still be applied to obtain the absolutely optimal strategy. The proposed algorithm might be seen as an approximation of it, which only requires solving a stage-wise minimax problem online instead of a global one offline.
4. There is a possibly interesting future direction, related to [Rakhlin et al., 2012] cited in Appendix A. The idea of [Rakhlin et al., 2012] is that we can first write down the conditional value function achieved by DP, then try to relax (upperbound) it by a tractable potential function. This argument has been improved recently [Drenska and Kohn, 2020; Kobzar et al., 2020], where potential functions are more easily obtained by solving a PDE. For the expert problem with few experts, this yields tighter bounds compared to EW. Also, it allows general terminal conditions: instead of $\max_j R^T_j$, we can bound $\phi(R^T_1,\ldots,R^T_d)$ for more general $\phi$. \
Not sure if this PDE approach extends to the setting of the present paper, but if it does, then the generalized terminal condition might also be useful in related applications, e.g., calibration, approachability, etc. \
\
Drenska, Nadejda, and Robert V. Kohn. "Prediction with expert advice: A PDE perspective." Journal of Nonlinear Science 30.1 (2020): 137-173.\
Kobzar, Vladimir A., Robert V. Kohn, and Zhilei Wang. "New potential-based bounds for prediction with expert advice." Conference on Learning Theory. PMLR, 2020.
5. Some minor suggestions on organization. Maybe it's just a personal preference, I feel the last two applications (No-regret expert & approachability) are easier to read compared to calibration, therefore might be moved forward (after Section 2) for better clarity. Section 3.2 (multicalibeating) is quite dense, so I suggest shortening it using high-level arguments rather than precise arguments (details are in the appendix anyway). In particular, the proof sketch of Theorem 3.2 and the statement of Theorem 3.3 (it's complete but really hard to parse).
6. Very minor:
- The setting of multicalibeating involves level sets (Line 240), but it's not defined until Line 275.
- The predictor for multicalibeating is denoted as $a$. Instead, $\mathbb{A}$ or $\mathcal{A}$ might be clearer?
- Line 91, $2^\theta\rightarrow 2^\Theta$.

**Limitations:**

Limitations are truthfully stated in the paper. This is a theoretical work, therefore the societal impact questions are not applicable.

**Strengths And Weaknesses:**

This is a strong submission in my opinion, and overall a pleasant read.

Strengths:
1. The setting of the framework is simple but general. The proposed solution has a EW flavor but differs in some substantial ways that I am not aware of in existing works. This could be interesting to a large subset of the community. In particular, the algorithm "upweights" the coordinates with higher historical losses in the resulting convex-concave problem, which is natural but insightful.
2. The paper provides an extensive discussion on existing works. Contributions are clear, and limitations are truthfully presented.
3. Technical extensions and applications are thoroughly developed, with clear proofs.
4. The paper is well-written in general. It is a dense paper, but the authors delivered the key idea quite clearly.

Weaknesses:
I don't have any major complaints on the paper; its current form is already good. Some minor suggestions are provided in the following.

---

> ### Author Response · Authors · 2022-08-01
> **Reply to reviewer mPvY**
>
>
> Thank you very much for your detailed and insightful review! Your suggestions for future directions and connections to the literature are very interesting, and we will think carefully about them. To address your specific points:
>
> 1. We agree with your comments on the technical and aesthetic connections to EW, and will include additional discussion in our next revision.
>
> 2. We agree it would be interesting to improve the constant to the minimax optimal one, and will think about ways to do so. It is worth mentioning that we stated the constant as 4 for the sake of simplicity, even though it can easily be further reduced (to at least below 3) with basically the same technique.
>
> 3. We fully agree with this interpretation --- one could view the method we propose as a greedy strategy for breaking up a global Rakhlin et al. style DP problem (min max min max ... over all rounds) into local computations (min max in each round).
>
> 4. This is an intriguing connection, thank you for the pointers to these papers. We will give it further thought and add a pointer to these papers and this future work direction in the revision.
>
> 5. Your suggestion for improving the exposition by moving the simpler applications up front is also well taken, and we plan to do this as we revise the paper. The reason we did not do this in the submission is that we wanted to have space to present some of the more novel applications, but we are in complete agreement with you about the ordering of the results best for exposition.
>
> 6. Thank you for spotting these typos, we will fix them.

---

> > ### Comment · Reviewer_mPvY · 2022-08-05
> > **Post rebuttal update**
> >
> > Thanks for the response. This is a good paper, and I will continue supporting acceptance. Congrats for the good work!

---

> > > ### Author Response · Authors · 2022-08-05
> > > **Thanks!**
> > >
> > > Thank you!

---

### Official Review · Reviewer_wRUm · 2022-07-12

**Rating:** 5
**Confidence:** 2
**Soundness:** 3 good
**Presentation:** 3 good
**Contribution:** 3 good

**Summary:**

This paper consider an online adversarial multiobjective minimax optimization scenario, where at each round t, the adversary chooses an environment defined by a convex compact action set $\mathcal{X}^t$ for the learner, a convex compact action set $\mathcal{Y}^t$, then learner choose action $x_t$ and adversary chooses action $y_t$. The d dimensional loss function is defined such that each of its coordinate is convex in x and concave in y. We use the maximum of these coordinate as the final loss, i.e., $\max_{j\in [d]}\ell_j^t(x_t,y_t)$. The authors proposed an online exponential multiplicative algorithm to solve this minimax problem and show that it achieve sub-linear regret bound, i.e., $\sqrt{T}$. They further show a direct application of their algorithm: Multicalibration and Multicalibeating.


**Questions:**

I think there are some previous works in Multicalibration and Multicalibeating, so can authors compare their algorithm with others, what is the advantage of their method?

**Limitations:**

I did not see negative societal impact of this work.

**Strengths And Weaknesses:**

This paper consider a novel online learning scenario, where the objective is the maximum coordinate of a vector loss, and borrow the idea from bandit learning to solve it. The proof looks good to me. They also give applications of their framework in fairness field. However, I am not familiar with fairness including Multicalibration and Multicalibeating, so I cannot judge the value of their application.

---

> ### Author Response · Authors · 2022-08-01
> **Reply to reviewer wRUm**
>
> Thank you for your review! Algorithms for multicalibration existed prior to our work, but were derived using a specialized analysis. The main contribution of our paper is to derive --- among other things --- multicalibration algorithms as a simple application of a new common framework that can also be used to recover a large variety of other algorithms (e.g., many families of no-regret algorithms, and fast Blackwell approachability methods).
>
> In addition to our new framework, our application to "multicalibeating" is new. Prior work on calibeating (by Foster and Hart) used different techniques and had error terms that depended polynomially on the number of models to be "calibeat". Our results give an exponentially improved dependence --- our algorithms have error terms depending only logarithmically on the number of models to be ``calibeaten''.  Our algorithms are also the first that can simultaneously ``calibeat'' many models while being multicalibrated, which we call ``multicalibeating.''

---

> > ### Comment · Reviewer_wRUm · 2022-08-09
> > **Thanks for authors' response**
> >
> > I would like to thank authors for their detailed explanation. I tend to keep my score.

---

### Meta-Review · Area_Chair_5Ayx · 2022-08-21

**Recommendation:** Accept
**Confidence:** Certain

**Metareview:**

There is general agreement that this paper should be accepted.

**Award:**

No

---

### Decision · Program_Chairs · 2022-09-14

Accept